## METHOD

# Normalization and de-noising of single-cell Hi-C data with BandNorm and scVI-3D

Ye Zheng[1†], Siqi Shen[2†] and Sündüz Keleş[2,3*] 

†Ye Zheng and Siqi Shen contributed equally to this work.

*Correspondence:
keles@stat.wisc.edu

[1] Vaccine and Infectious Disease Division, Fred Hutchinson Cancer Center, Seattle, USA
[2] Department of Biostatistics and Medical Informatics, University of Wisconsin - Madison, Madison, USA
[3] Department of Statistics, University of Wisconsin - Madison, Madison, USA

## Abstract

Single-cell high-throughput chromatin conformation capture methodologies (scHi-C) enable profiling of long-range genomic interactions. However, data from these technologies are prone to technical noise and biases that hinder downstream analysis. We develop a normalization approach, BandNorm, and a deep generative modeling framework, scVI-3D, to account for scHi-C specific biases. In benchmarking experiments, BandNorm yields leading performances in a time and memory efficient manner for cell-type separation, identification of interacting loci, and recovery of cell-type relationships, while scVI-3D exhibits advantages for rare cell types and under high sparsity scenarios. Application of BandNorm coupled with gene-associating domain analysis reveals scRNA-seq validated sub-cell type identification.

**Keywords:** Single-cell Hi-C, Normalization and de-noising, Cell type separation, Compartment and domain recovery, Gene associating domains, 3D genome marker genes

## Background

Maturation of chromosome conformation capture (3C)-based technologies for profiling 3D genome organization [1–5] and technological advancements in single-cell sequencing [6] led to the development of single-cell Hi-C (scHi-C) assays [7–12]. Data from these assays enhance our ability to study the impact of spatial genome interactions on cell regulation at an unprecedented resolution. While some of the characteristics of scHi-C data, such as systematic genomic distance bias (referred to as *band bias*, Fig. 1a), are similar to its bulk version, scHi-C data harbors significantly distinct features. In general, data from single-cell technologies such as scRNA-seq and scATAC-seq are noisy and sparse, leading to underestimated biological signals within and across cells. However, these issues are compounded in 3C-based technologies because the natural analysis unit is locus pairs depicting potentially interacting genomic loci; as a result, their sheer number exacerbates the sparsity. In contrast to bulk Hi-C data, which is available in small numbers of replicates owing to high sequencing cost, scHi-C is generated across thousands of cells simultaneously, therefore significantly increasing the resolution to

**Fig. 1** Benchmarking of scHi-C normalization and de-noising methods for cell-type clustering. **a** Band transformation separates scHi-C contact matrices into band-specific cell × locus pair matrices before conducting BandNorm normalization on each band matrix per chromosome per cell. BandNorm normalizes the raw interaction counts $Y_r^{cv}$ for locus pair $r$ in band $v$ and cell $c$ into normalized count $C_r^{cv}$. **b** Deep generative model, scVI-3D, for a single band matrix $v$ with entries in the form of locus pairs $r$ ($r \in \mathcal{A}(v)$) by cells $c$ ($c = 1, ..., N$). The raw interaction counts $Y_r^{cv}$ are modeled as a function of low-dimensional latent variables $\mathbf{z}_{cv}$. Refer to the Section 5 for a detailed mathematical introduction and practical settings of the scVI-3D model. **c** Evaluation of the eight scHi-C normalization and de-noising methods, namely CellScale, BandScale, BandNorm, scHiCluster, scHiC Topics, Higashi, CellScale+CNN, and scVI-3D, for cell type separation across four benchmark datasets. The performances are evaluated by Adjusted Rand Index (ARI) after K-means clustering and Louvain graph clustering and by Silhouette coefficient on UMAP and t-SNE visualizations with the true cell labels. **d** Median ranks of the performance of the scHi-C methods across the six evaluation metrics and four data sets

capture biological variation. However, this resolution gain comes at the cost of increased technical noise and decreased sequencing depth per cell, further contributing to the extreme sparseness of scHi-C chromosomal contact matrices.

Initial approaches for unsupervised analysis of scHi-C data repurposed bulk data quantification methods of similarity between contact matrices, such as HiCRep [13], and applied multi-dimensional scaling [14]. Vectorization of scHi-C contact matrices to form a cell by locus pair matrix followed by dimension reduction approaches such as UMAP and t-SNE [10] or topic modeling [15] have been utilized successfully. Most recent approaches for normalization and de-noising of scHi-C data rely on linear smoothing and random walk imputation [16] of cell-specific contact matrices or hypergraph representation learning [17]. While these are highly innovative approaches, they lack a generative model that acknowledges the key properties of the scHi-C data. Deep generative modeling and, more specifically, variational autoencoders have seen a significant uptake in the analysis of single-cell transcriptomics [18, 19], epigenomics [20], proteomics [21], and bulk 3D genomics [22] due to their ability to provide efficient and scalable solutions to normalize, de-noise, and impute single-cell data. At the individual

cell resolution, heterogeneity, driven by the stochastic nature of chromatin fiber and a multitude of nuclear processes, and unwanted variation due to sequencing depths and batch effects pose major analytical challenges for inferring single cell-level 3D genome organizations. Therefore, we first describe a computationally fast scaling normalization approach, named BandNorm (Fig. 1a), that operates on the stratified off-diagonals (i.e., bands) of the contact matrix and its variants as fast baseline alternatives, namely CellScale and BandScale, which have been utilized for bulk Hi-C and have seen some uptake for scHi-C [10, 23, 24] (Section 5). Next, we develop a deep generative model named scVI-3D, which systematically takes into account the structural properties and accounts for genomic distance bias, sequencing depth effect, zero inflation, sparsity impact, and batch effects of scHi-C data (Fig. 1b). Finally, we explore a single-cell gene associating domains (scGAD) score analysis to biologically interpret refined 3D genome clustering results at the gene level.

## Results

### Benchmark datasets

We leveraged four scHi-C datasets with known cell-type labels and varying characteristics to evaluate the performances of the newly proposed scHi-C normalization methods, BandNorm and scVI-3D, in comparison with the existing state-of-the-art approaches (Section 5). These four datasets are *Ramani2017* [8] and *Kim2020* [15] that include multiple human cell lines, *Lee2019* [10] that profiles human brain prefrontal cortex cells, and *Li2019* [9] of mouse embryonic stem cells (mESC). Of these, *Ramani2017* does not exhibit batch effects and the experimental design in *Kim2020*, where each cell type is either in a single batch alone or together with another cell line in two batches, largely confounds batches and cell types. In contrast, *Lee2019* is generated from 5 sequencing libraries, which exhibit explicit batch effects. Therefore, we leveraged *Lee2019* to investigate the impact of batch effects in detail. All the scaling-based normalization methods and methods that do not have built-in batch correction are coupled with Harmony [25], which has established effectiveness for scRNA-seq [26] and performed well in our experiments (Section 2.5), as a batch correction method unless otherwise stated. Data on chromosomes 1-22 and X are binned at 1Mb resolution to generate a set of loci, and raw data are filtered according to the specifications in the source publications to remove extremely sparse cells (Table 1 and Additional file 1: Fig. S1). *Lee2019* is additionally binned at 100kb to assess the performance of methods at a high resolution and high sparsity setting. Unless otherwise stated, 1Mb is the default resolution for the downstream analysis.

### Band transformation for BandNorm and scVI-3D

scHi-C contact map is a symmetric loci by loci matrix with each entry representing the interaction frequency between the locus pairs that are potentially in spatial proximity. The diagonal or each off-diagonal on the contact matrix is considered as a band, and the band transformation [23, 24] is the foundation for the BandNorm and scVI-3D normalization and modeling (Fig. 1a). The genomic distance effect, i.e., band effect, due to the random polymer looping behavior of DNA is one of the key features in both the bulk [1] and single-cell Hi-C data (Additional file 1: Fig. S2). As expected, such a band effect

**Table 1** Summary of single-cell Hi-C data for benchmark analysis

| Data source | Cell type | # of cell | Interaction frequency | Off–diagonal interaction frequency | # of locus pairs | # of off–diag locus pairs | Mean off–diagonal interaction frequency per cell | Mean # of off–diagonal locus pairs per cell |
|---|---|---|---|---|---|---|---|---|
| Ramani2017 [8] | GM12878, HAP1, Hela, K562 | 2610 | 51,420,688 | 23,089,064 (44.9%) | 16,639,155 | 11,081,243 (66.6%) | 8846.38 | 4245.69 |
| Lee2019 [10] | Astro, Endo, L2/3, L4, L5, L6, MG, MP, Ndnf, ODC, OPC, Pvalb, Sst, Vip | 4238 | 4,586,887,008 | 553,896,738 (12.08%) | 132,310,725 | 120,126,414 (90.79%) | 130,821.2 | 28,371.85 |
| Li2019 [9] | 2i; Serum1, Serum2 | 150 | 15,846,298 | 2,612,886 (16.49%) | 1,906,405 | 1,523,307 (79.9%) | 17,419.24 | 10,155.38 |
| Kim2020 [15] | GM12878, H1Esc, HAP1, HFF, IMR90 | 9230 | 95,764,991 | 39,702,592 (41.46%) | 48,866,268 | 30,333,591 (62.07%) | 4305.2 | 3289.26 |

leads to marked interaction frequency enrichment among loci close to the diagonal in the Hi-C contact matrix. Contact decay profiles that quantify interaction frequencies among locus pairs as a function of their genomic distance can successfully separate cells based on their cell cycle stages [27]. To explicitly capture this effect, the upper triangular of the symmetric contact matrix for each cell is first stratified into diagonal bands, each representing a specific genomic distance between the interacting loci. Then, bands at the same genomic distance are combined into a band matrix across cells (Fig. 1a) for further BandNorm and scVI-3D normalization (Section 5). For instance, each purple off-diagonal within a cell in Fig. 1a forms a column in the purple locus pair by cell matrix. Together, all the purple off-diagonals across cells construct the purple band matrix, and all the locus pairs in such a band matrix share the same genomic distance effect. By dividing the interaction frequencies of each band within a cell with the cell-specific band mean, BandNorm first removes genomic distance bias within a cell and scales the sequencing depths between cells. Subsequently, BandNorm adds back a common band-dependent contact decay estimate by multiplying each band within a cell with the average band mean across cells (Fig. 1a and Section 5). A similar strategy for imposing band-specific decay rates was adopted for normalization across bulk Hi-C samples [28].

In addition to this nonparametric normalization approach, we also devised scVI-3D as a deep generative model built on the parametric count models of Poisson and Negative Binomial that have been successfully used in bulk measurements of chromatin conformation capture data [29, 30]. Motivated by the recent deep learning modeling approaches for single-cell transcription [18, 19] and chromatin accessibility [20], scVI-3D builds a generative modeling framework on the band matrices for dimension reduction and de-noising explicitly designed for the scHi-C data (Fig. 1b and Section 5). scVI-3D estimates and corrects the sequencing depth and potential batch effect biases and de-noises interaction frequencies among locus pairs that can then be leveraged for downstream analysis. Specifically, scVI-3D keeps all the features (i.e., locus pairs) instead of extensively discarding low variable features as routinely done in scRNA-seq and scATAC-seq analysis. However, cells with no interaction across all the locus pairs within the target band are filtered before the variational inference for estimating model parameters. The resulting latent embeddings for such zero interaction cells are imputed by 0 before concatenating the latent embeddings across bands while matching the cell identity. scVI-3D also adapts a progressive pooling strategy [23]. This pooling strategy significantly improves the cell type separation performance and robustifies the normalization process, particularly for the distant off-diagonal bands, which tend to have fewer features and severe sparsity (Additional file 1: Fig. S3). The dimension of the latent space also influences the scVI-3D performance, and the commonly used latent variable dimension for scRNA-seq or scATAC-seq (i.e., 10–50) is typically not appropriate for scHi-C data. Our empirical investigations suggest that setting the default latent variable dimension of scVI-3D to 100 can generally achieve a good performance in separating the major cell types (Additional file 1: Fig. S4). Additionally, the zero-inflation component enables scVI-3D to better fit at high resolutions and, hence, for high sparsity scHi-C data settings. This alone exceeds the modeling capacity of the baseline scaling-based normalization methods (Additional file 1: Fig. S5).

BandNorm is implemented as an R package (https://github.com/keleslab/BandNorm) which also harbors all the curated public scHi-C data used in this paper. Expanding the scvi-tools [18], scVI-3D is implemented as a python pipeline and is available at https://github.com/keleslab/scVI-3D.

### Existing scHi-C methods

We benchmarked BandNorm and scVI-3D against two classes of methods, including baseline methods for library size and genomic distance effect normalization and more structured modeling approaches. In the former category, in addition to BandNorm, we devised and evaluated two more baseline scaling-based normalization methods: CellScale and BandScale [10, 23, 24] (Section 5). CellScale uses a single scaling factor across all the locus pairs within a cell, while BandScale employs a band-specific size factor, as opposed to a global one, within each cell to eliminate library size bias at each genomic distance [10]. After each of CellScale, BandScale, and BandNorm normalizations, single-cell contact matrices are vectorized into the cell by locus pair matrices and used to generate low-dimensional embeddings. Since this strategy overlooks the matrix structure of the data, we also utilized a convolutional neural network (CNN) approach, which has been previously leveraged for enhancing the resolution of the bulk-cell Hi-C matrix [31]. CNN method can be coupled with contact matrices pre-processed by either of the three baseline methods, CellScale, BandScale, and BandNorm, to learn their low-dimensional representations. CellScale+CNN stands out after systematic comparisons on *Ramani2017*, *Lee2019*, and *Kim2020* data sets quantitatively and visually regarding the cell type separation (Additional file 1: Fig. S6). Therefore, for the benchmarking of the downstream analysis, only the CellScale+CNN strategy is utilized. In the second category, we considered three existing state-of-the-art scHi-C methods, namely, scHiCluster [16], scHiC Topics [15], and Higashi [17].

### Impact on cell type separation

We first assessed the cell-type separation performances of the scHi-C methods within the context of unsupervised clustering. We considered six evaluation settings (Fig. 1c, Additional file 1: Fig. S7 and Section 5): K-means clustering of the latent embeddings of each method, and with or without low-dimensional projections with t-SNE and UMAP, and Louvain graph clustering [32]. We used Adjusted Rand Index (ARI) to compare the resulting clusters with the known cell labels and the average Silhouette score to quantify the separation between the clusters. In addition to these quantitative measures, we also graphically assessed whether the resulting low-dimensional representations achieve clear cell type separation (Figs. 2 and 3a, b) and carry left-over effects by technical variation due to batch (Fig. 3c-f), library size or sparsity (Additional file 1: Fig. S8). We present *Ramani2017*, *Lee2019*, and *Kim2020* with UMAP visualizations for illustration (Figs. 2 and 3a-b) and provide the t-SNE embeddings (Additional file 1: Figs. S9-S11) as well as analysis for *Li2019* (Additional file 1: Figs. S12 and S13) in the supplementary information.

Cell clustering performances vary dramatically across data sets due to the numbers of cells, data quality measured by sequencing depth, sparsity, and batch effects, as well as

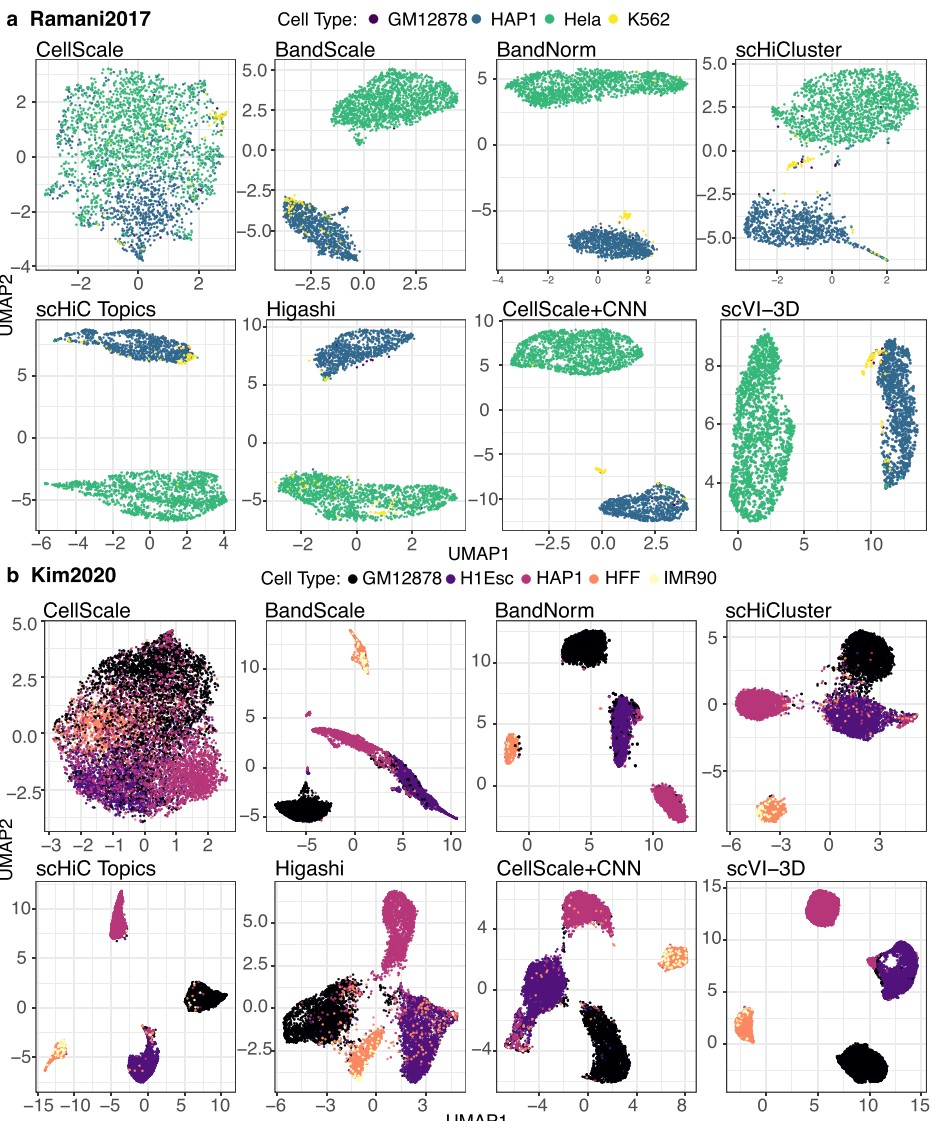

**Fig. 2** Comparison of scHi-C normalization and de-noising methods for their performances in separating cells from major cell lines. Application of the scHi-C data normalization and de-noising methods on *Ramani2017* (**a**) with 4 cell types and 2610 cells and *Kim2020* (**b**) data sets with 5 cell types and 9230 cells. The results are displayed using scatter plots of the first two UMAP coordinates. The colors of the plotting symbols depict the cell types

how distinguishable the cell types are. For the *Ramani2017* and *Lee2019* studies, Band-Norm and scVI-3D perform as well as or outperform the rest based on all the six evaluation settings (Fig. 1c and Additional file 1: Fig. S7). Almost all the methods perform poorly on the *Li2019* dataset (ARI $\in [0.005, 0.47]$ and Silhouette score $\in [-0.56, 0.2]$), which has the smallest number of cells (Table 1 and Additional file 1: Fig. S1). All the methods except CellScale achieve their best performances on the *Kim2020* dataset (ARI $\in [0.35, 0.91]$ and Silhouette score $\in [0.3, 0.73]$), with leading performances by Band-Norm, scVI-3D, and scHiC Topics (Fig. 1c and Additional file 1: Fig. S7). Overall summary of these evaluations yields BandNorm and scVI-3D as robustly best performing, followed by Higashi and scHiCluster, with median rank scores of 2, 3, 3.5, and 4, respectively (Fig. 1d and Additional file 1: Fig. S7).

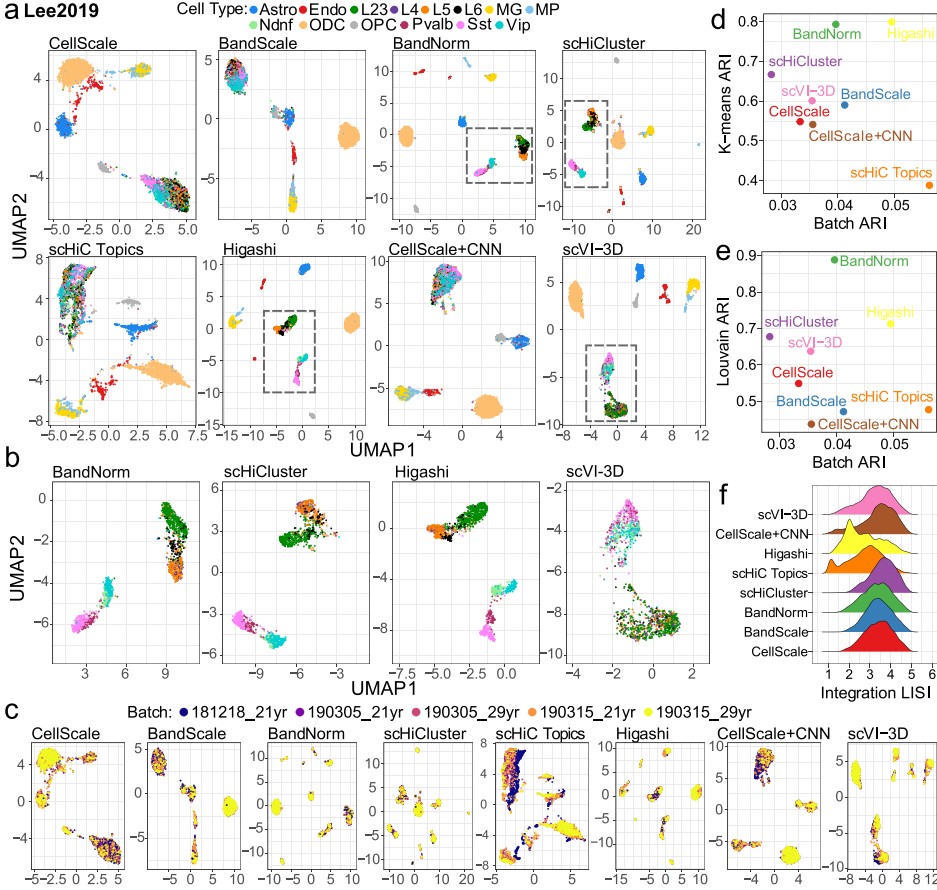

**Fig. 3** Separation of the neuronal sub-cell types in the *Lee2019* data set and the impact of the batch effects. **a** Application of the scHi-C data normalization and de-noising methods on *Lee2019* data set with 14 neuronal cell types. The results are displayed using scatter plots of the two UMAP coordinates. The colors of the plotting symbols correspond to the cell types. Excitatory neuronal subtypes (L2/3, L4, L5, L6) and inhibitory cells (Ndnf, Vip, Pvalb, and Sst) are highlighted by the gray dashed squares, which are amplified in **b** for the four leading methods: BandNorm, scHiCluster, Higashi, and scVI-3D, respectively. **c** Impact of batch effects on cell type separation using *Lee2019* data set with samples from two donors of ages 21 and 29 years old and in a total of 5 batches. The results are displayed using scatter plots of the two UMAP coordinates. The colors of the plotting symbols correspond to the batches. **d, e** Cell type separation and batch effect removal performances of the methods, evaluated by ARI with K-means clustering ($k =$ number of cell types) in **d** and Louvain clustering in **e**. ARI assesses the batch effect with K-means clustering ($k =$ number of batches). **f** Density of integration local inverse Simpson's Index (iLISI) scores [25] of cells evaluating the batch mixing performance after normalization by each of the eight methods. A high density around 1 indicates only one batch in a particular cell's neighborhood, demonstrating the batch effect. Larger values indicate well mixing across the batches

Next, we highlight some observations regarding the performances of each individual method. CellScale is a commonly used global library size scaling strategy for high-throughput sequencing data but produces the least favorable cell separation. Particularly, for *Ramani2017* and *Kim2020* (Fig. 2), where most of the other methods can explicitly separate major cell types, CellScale exhibits no separation power, which is consistent with the ARI and Silhouette scores evaluation (Additional file 1: Fig. S7). Furthermore, BandScale and CellScale+CNN achieved a slightly better cell type separation than CellScale. A direct comparison of the results from BandScale to BandNorm highlights the marked improvement due to adding back the band-specific contact decay estimates

by BandNorm (Figs. 2 and 3a, b, and Section 5). Allowing band-specific contact decays can be interpreted as adding more weights to short-range bands, which play a significant role in separating the cell types. While CellScale+CNN strategy explicitly acknowledges the matrix structure of the data with a potential to learn the graph structure of each cell matrix, it yields limited power, possibly owing to the sparsity and low resolution of 1Mb. While scHiC Topics performs impressively well on *Kim2020*, it does not exhibit similarly high performance on the other datasets and is particularly challenged in the *Lee2019* data set. This is in spite of the fact that we used the suggested strategy [15] of setting the number of topics based on the true cell labels in the Silhouette score calculations. The rest of the four methods, BandNorm, scVI-3D, Higashi, and scHiCluster, achieved generally high scores in separating cell types from the four data sets with particularly outstanding performances in distinguishing the 14 cell types of human brain prefrontal cortex in *Lee2019* (Fig. 3a). The original analysis of *Lee2019* reported that scHi-C data alone could barely separate the excitatory neuronal subtypes, namely L2/3, L4, L5, and L6, from the inhibitory cells Ndnf, Pvalb, Sst, and Vip. While this is the case for other baseline methods, BandNorm, scHiCluster, and Higashi show a notable exception, unambiguously separating the excitatory and inhibitory cells into two clusters (Fig. 3b). Consistent with this, scVI-3D also achieves segregation between the excitatory and inhibitory cells at 1Mb resolution (Fig. 3b) and even further refined separation across the sub-cell types of excitatory neurons and inhibitory cells at 100kb resolution (Additional file 1: Fig. S5). For this analysis of the *Lee2019* dataset, as indicated previously, methods without batch correction, namely CellScale, BandScale, BandNorm, CellScale+CNN, scHiC Topics, were coupled with Harmony [25] for batch correction, and the other methods leveraged their built-in batch correction. Next, we utilized the single largest library of *Lee2019* to create a batch-free context for additional systematic comparison. Consistent with findings using the full *Lee2019* data, BandNorm, Higashi, scHiCluster, and scVI-3D achieved clear cell type separation for the major clusters (Additional file 1: Fig. S14), further suggesting that their better performances are not likely due to better batch correction.

The relatively poor performances of scHiCluster and Higashi on the *Kim2020* dataset unearth specific issues. scHiCluster requires stringent cell filtering (Table 2 and Section 5) that may remove 88.4% of cells in datasets such as *Kim2020* [15]. Without such stringent filtering, scHiCluster loses its cell type separation power unexpectedly. In contrast, in the *Kim2020* study with all the 9230 cells from the source paper [15], BandNorm and other more structured models, such as scHiC Topics and scVI-3D, achieve distinct separation with high ARI and Silhouette scores (Fig. 2b). scHiCluster, however, performs poorly in distinguishing cells with low sequencing depth and high sparsity, leading to the mixing of cells from different cell types (Fig. 2b and Additional file 1: Fig. S11). This is consistent with the observations of others [15, 17] on scHiCluster. The most recent version of the Higashi software successfully resolved the batch effect issues and exhibited the leading performance in the separation of the sub-neuronal and inhibitory cells at both the 1Mb and 100kb resolutions (Fig. 3b and Additional file 1: Fig. S5). However, the issues with distinguishing cell types of cells with relatively smaller library sizes and higher sparsity from the *Kim2020* study, which was a bottleneck for scHiCluster, became exacerbated for Higashi. This led to inaccurate cell clustering with lower ARI and Silhouette scores (Figs. 1c and 2b and Additional file 1: Fig. S7).

**Table 2** Overview of single-cell Hi-C data analysis methods

| Method | Resolution | Cell filtering | Locus pairs filtering | Sequencing depth normalization* | Batch effect removal | Time (Lee2019 1Mb) | Memory (Lee2019 1Mb) |
|---|---|---|---|---|---|---|---|
| CellScale | 1Mb | NULL | NULL | Each locus pair is normalized by dividing with the total IFs of each contact matrix and multiplying by 10,000 | NULL | < 15 min (23 cores) | 20G |
| BandScale [10] | 100kb | NULL | Keep < = 2Mb | Each locus pair is normalized by dividing with the average IFs of locus pairs at the same distance | NULL | < 15 min (23cores) | 20G |
| **BandNorm** | 1Mb | NULL | NULL | Each locus pair is normalized by dividing with the total IFs at the same genomic distance (i.e., band) per cell and multiplying by the average total interaction frequency across cells at the same distance | Top batch correlated PCA components other than the 1st and 2nd are removed | **<15 min(1 core)** | 20G |
| scHiCluster [16] | 1Mb 200kb | Keep cell with off–diagonal contacts > 5000. | Keep cells with > = x non–zero locus pairs, x is chromosome size (Mb) | After neighborhood smoothing and random walk imputation, top 20% interacting locus pairs are set to 1 and rest to 0 | NULL | 8 h (23 cores) | 204G |
| scHiCTopics [15] | 500kb | NULL | Keep < = 10Mb | Cell by locus pair matrix is binarized; (optional) bottom 50% cell based on total IFs are removed and the rest is downsampled to remove the sequencing depth effect | NULL | 36 h (1 core) | 14.6G |
| Higashi [17] | 1Mb | Keep cells with >= 2,000 interactions that have genomic span greater than 500Kb. | NULL | Contact maps can be smoothed via linear convolution to reach similar sparsity; quantile normalization can be applied to the matrix | NULL | 24 h (23 core CPU) 1 h (23 cores GPU) | 253G (CPU) 110G (GPU) |
| CellScale+CNN | 1Mb | NULL | Keep cells with > = x/6 non–zero locus pairs, x is chromosome size (Mb) | Each locus pair is normalized by dividing with the average IFs of locus pairs at the same distance. | NULL | 5 h (23 cores) | 92G |
| **scVI–3D** | 1Mb | NULL | (Optional) Diagonals of contact matrix with high contact count variation are selected | A variational autoencoder with Gamma–Poisson mixture at each band with a separate size factor is fitted | The batch factor is incorporated into the autoencoder at each distance | **20 min (23 core GPU)** | 110G (GPU) |

BandNorm achieves the most stably high performance across data sets in separating the cell types. Remarkably, the BandNorm normalization is ultra-fast (15 mins compared to hours or even more than a day for other methods, Table 2) and requires a small amount of memory. Fast processing and robust performance make BandNorm particularly suitable for rapidly diagnosing data quality and cell type separation on scHi-C datasets or pilot studies. scVI-3D is generally performing as well as BandNorm in terms of separating cell types, and its overall high performance is stable across data sets (Fig. 1c and Additional file 1: Fig. S7).

### Impact of batch effects

Batch effects are known to modulate cell type separation performances of different methods in other single-cell applications [25, 33]. scVI-3D explicitly incorporates batch factors into the deep generative framework and exhibits improved cell type separation and reduced batch impact (Additional file 1: Fig. S15). scHiCluster and Higashi implementations also have built-in batch effect correction components. While BandNorm already achieved generally high cell-type separation without batch correction (Additional file 1: Fig. S16A-B), consideration of the distance effect and library size without any adjustment for the batches exhibited an unexpected cell separation within excitatory neuronal subtypes of *Lee2019* (L2/3, L4, L5, L6, Additional file 1: Fig. S16A-"None" panel). To enable batch correction for BandNorm, we considered several methods that yielded promising results in removing batch effects for other types of high-throughput sequencing data, including SVA [33], removing the principal component exhibiting the highest correlation with the batch variable, Seurat batch effect regression [34], and Harmony [25]. Of these, SVA [33] and removing the principal component exhibiting the highest correlation with the batch variable did not eliminate the batch bias, while SVA even worsened the performance (Additional file 1: Fig. S16A). Seurat batch effect regression [34] alleviated the separation within the excitatory neuronal subtypes cluster but introduced additional batch biases for other clusters (depicted in orange and yellow in Additional file 1: Fig. S16A, especially for Astro, MG, and ODC cell types). In contrast, Harmony [25] stood out in successfully addressing the batch effects and slightly enhancing the sub-cell type separation for the excitatory sub-neurons and inhibitory cells (Additional file 1: Fig. S16B-C). Therefore, we coupled Band-Norm with the Harmony [25] batch correction for settings with batch effects. In addition, we also coupled Harmony with all the methods that do not explicitly model batch factors, including CellScale, BandScale, scHiC Topics, and CellScale+CNN, in our benchmarking experiments. Visualization based on low-dimensional embeddings of the methods and the quantitative evaluation on the remaining batch effect using ARI and integration local inverse Simpson's Index (iLISI) scores [25] indicates scHiC Topics as still slightly affected. The rest of the methods appear to be adequately handling the batch effects (Fig. 3c–f).

In addition to confounding cell-type separation, batch effects may also impact the inference of cell type relationships from the pairwise similarity measurements of the cells (Fig. 4). We compared the robustness of the methods regarding this effect by considering a cell-to-cell similarity metric based on the edge weights of shared nearest neighbor graphs [35], which we found to properly capture the similarity of two contact matrices at the single-cell resolution. As revealed by the hierarchical clustering of the cells based on this metric (Fig. 4 and Additional file 1: Fig. S17), CellScale, BandNorm, scHiCluster, scHiC Topics, Higashi, and scVI-3D latent embeddings

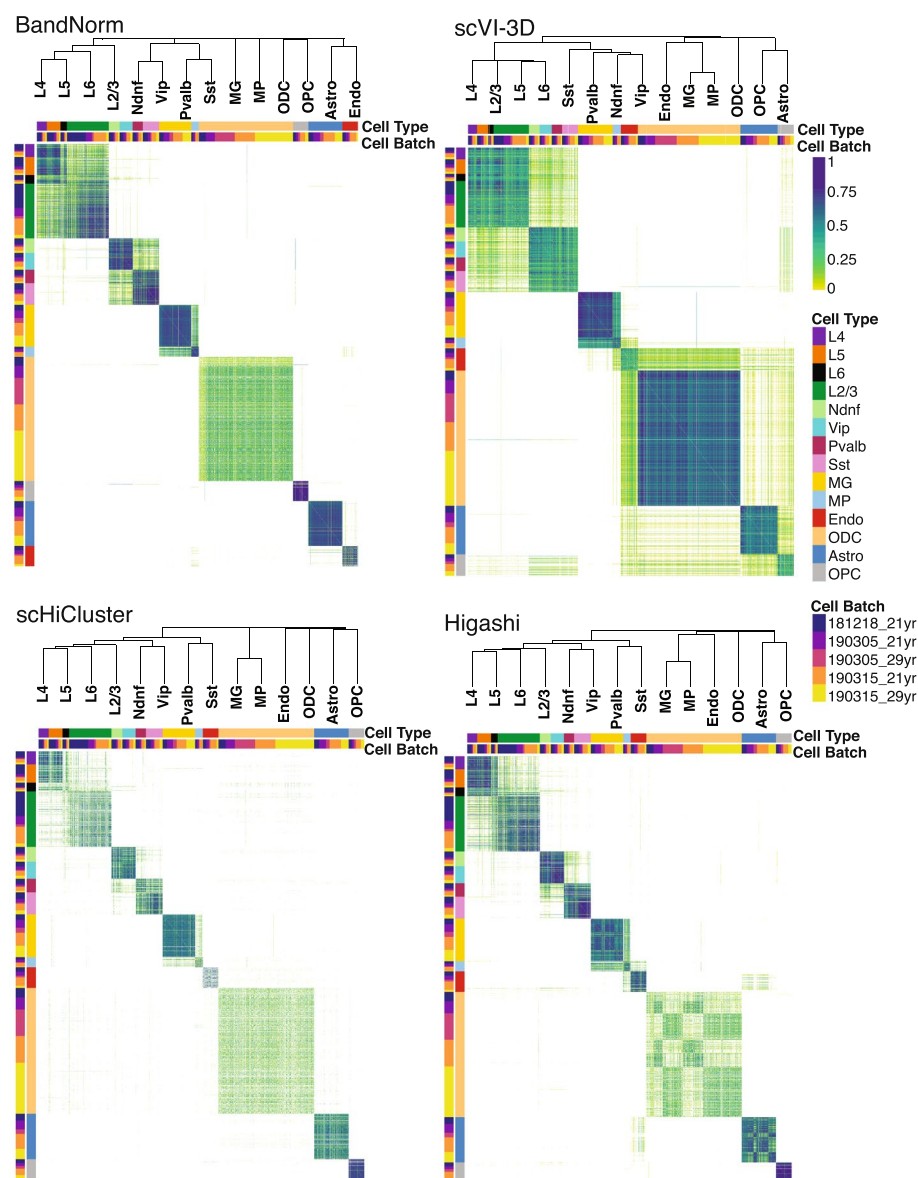

**Fig. 4** Cell-cell similarity analysis to recover cell type relationships. Pairwise cell similarity scores for the *Lee2019* data set are obtained by edge weights of the shared nearest neighbor graphs constructed from normalized data (low-dimensional embeddings from scVI-3D, scHiCluster, and Higashi, and the first 50 PCs for BandNorm). The inferred relationships between the cell types are depicted by the dendrogram from the hierarchical clustering of the cells

successfully separate the excitatory neuron subtypes (L2/3, L4, L5, and L6), CGE-derived inhibitory subtypes (Ndnf and Vip), medial ganglionic eminence-derived inhibitory subtypes (Pvalb and Sst), oligodendrocyte related cell types (Astro, OPC, ODC) and non-neuronal cell types (MG, MP, Endo). BandNorm coupled with Harmony batch removal, scHiCluster, and scVI-3D yield the most invariant performance to the batch effects on the similarity matrix (Fig. 4 and Additional file 1: Fig. S17).

### Impact of sequencing depth and sparsity

Another notable observation from our cell type separation benchmarking study is that UMAP and t-SNE visualizations of low-dimensional embeddings from Band-Scale, scHiCluster, scHiC Topics, Higashi, CellScale+CNN, and scVI-3D tend to display systematic effects of sequencing depth and sparsity despite their implicit or explicit efforts to account for these factors (Additional file 1: Fig. S8). However, these lingering effects do not impact the overall cell type separation. Instead, they tend to impact the local organization of the cells within their respective clusters. We considered utilizing PCA on the low-dimensional embeddings (t-SNE or UMAP) to identify and remove the top principal component(s) highly correlated with sequencing depth and sparsity. However, their removal did not result in any discernible improvement in the cell type separation and, on the contrary, led to worse evaluation metrics for some datasets. This eludes the possibility that the sequencing depth and sparsity effects left in the model embeddings are confounded with other latent biological or technical variations and cannot be completely removed from the data.

### Impact of normalization and de-noising on downstream analysis

Next, we sought to evaluate the normalization and de-noising by BandNorm and scVI-3D along with other methods with leading performances for their impact on downstream scHi-C data analysis, leveraging the *Kim2020* data set as an illustration. We compared aggregated scHi-C contact matrices of individual cell types after normalization (BandNorm) or de-noising (scVI-3D, Higashi, and scHiCluster) with their existing bulk Hi-C versions as the gold standard in terms of detection of A/B compartments and topologically associating domains (TADs; Figs. 5 and 6, Additional file 1: Figs. S18-S19), contact matrix similarity and detection of significantly interacting (Fig. 7 and Additional file 1: Fig. S20), as well as the differentially interacting locus pairs (Additional file 1: Fig. S21). The results presented are for scHi-C aggregation based on the true cell type labels; however, the overall benchmarking conclusions remain the same for the aggregation using unsupervised clustering labels (Section 5). When the number of cells per cell type is large such as in GM12878 (Additional file 1: Fig. S1), BandNorm normalized and aggregated scHi-C data exhibits good visual agreement with the bulk version in terms of the broad features of the contact matrix, such as the TADs and A/B compartments (Fig. 5). However, for rare cell types, i.e., IMR90 in the *Kim2020*, data from BandNorm exhibits extreme sparsity and does not yield good concordance with the bulk version (Fig. 6). In contrast, contact matrices de-noised with scVI-3D appear more in agreement with their bulk version regardless of the number of cells (Figs. 5 and 6). Higashi results in a blurry pattern on the aggregated contact matrix, obscuring the boundaries of compartments and domains. This can be potentially attributed to smoothing across the cells in the hypergraph setting. Additionally, aggregated scHi-C data from Higashi consistently yields unexpected over-enriched locus pair clusters in the off-diagonal regions across all five cell types (black arrows in Figs. 5 and 6, and Additional file 1: Figs. S18-S19). This indicates a potential systematic over-imputation issue driven by outlier cells with artifacts in the long-range interaction regions (Additional file 1: Fig. S22). scHiCluster de-noises the scHi-C contact matrix through neighborhood smoothing and random walk;

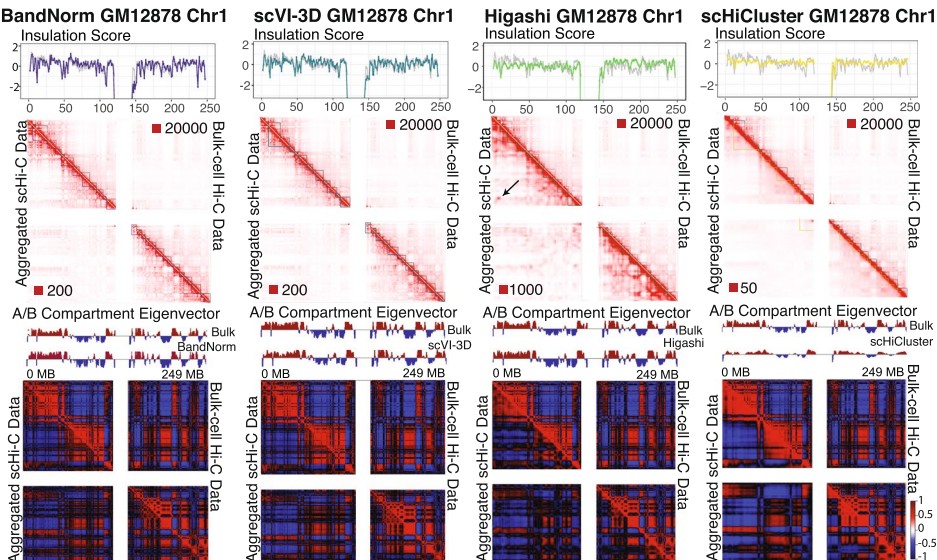

**Fig. 5** Impact of scHi-C normalization and de-noising methods on the compartment and domain detection for GM12878. Comparison of detected topologically associating domains (TADs) and A/B compartments between bulk GM12878 Hi-C data on chromosome 1 (upper right triangles) and the aggregated single-cell Hi-C data (lower left triangles) after normalization or de-noising using *Kim2020* data set. The numbers after the red squares at the left bottom or right upper corner of each contact matrix represent the minimum interaction frequency for the reddest locus pair. The black arrow highlights one example region that keeps showing over-imputation artifacts by Higashi across all five cell types compared to the bulk Hi-C data as a gold standard. True cell labels for GM12878 were utilized to aggregate the processed scHi-C data. First row: The insulation scores [36] that trace the TAD boundaries are depicted on the contact matrices with gray lines corresponding to bulk Hi-C data and purple for BandNorm, blue for scVI-3D, green for Higashi, and yellow for scHiCluster. Second row: A/B compartments are detected using the eigenvector of the correlation map of bulk (upper right triangles) or aggregated (lower left triangles) Hi-C matrices, values of which are displayed above each correlation matrix

hence, the matrices look even more smooth and blurry than Higashi in comparison with their bulk versions.

Systematic quantification of these observations indicates their generality and consistency across the cell types. At the domain structure level, despite the sparsity in the aggregated scHi-C matrices, BandNorm has the most concordant insulation score (purple lines in the "Insulation Score" panel of Figs. 5 and 6, Additional file 1: Figs. S18-S19) to that of bulk data (gray lines in the "Insulation Score" panel of Figs. 5 and 6, Additional file 1: Figs. S18-S19). This results in the highest recovery rate for TAD boundaries across all the chromosomes and all the five cell types (Fig. 7a, e.g., the median recovery rates for GM12878 are 85.71%, 66.67%, 60% and 60.87% for Band-Norm, scVI-3D, Higashi and scHiCluster, respectively). Furthermore, HiCRep [13] similarity analysis confirms that BandNorm normalization has the overall highest reproducibility score with the bulk Hi-C data, followed by scVI-3D and Higashi (Fig. 7b). Additionally, pairwise cell line reproducibility analysis reveals that Higashi and scHiCluster consistently have higher similarity scores between IMR90 and any of the rest cell types compared to the bulk version as a reference level (Fig. 7c). This can be partially attributed to Higashi's imputation strategy that borrows information from neighboring cells and scHiCluster's smoothing and random walk around neighboring

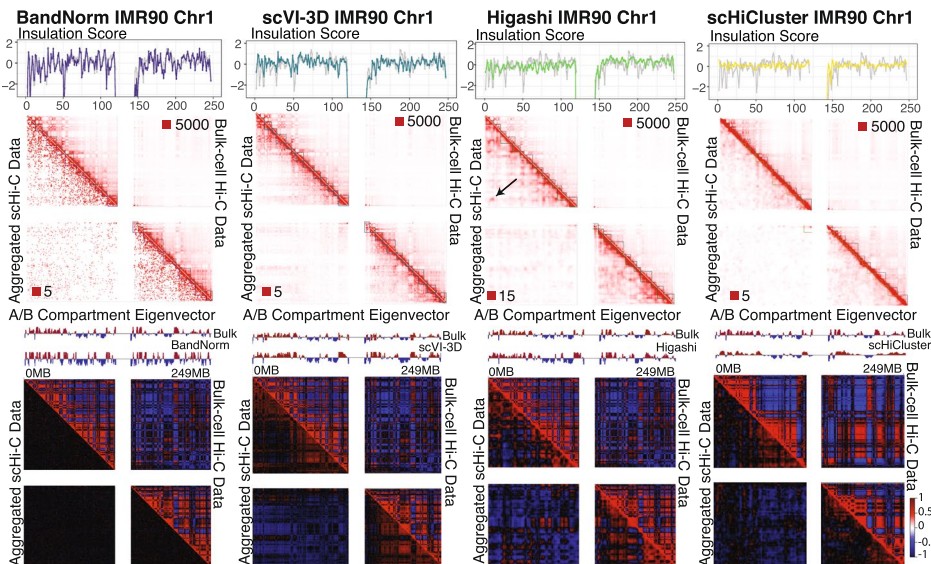

**Fig. 6** Impact of scHi-C normalization and de-noising methods on the compartment and domain detection for IMR90. Comparison of detected TADs and A/B compartments between bulk IMR90 Hi-C data and the aggregated single-cell Hi-C data after normalization or de-noising using *Kim2020* data set with the known IMR90 cell type label. The numbers after the red squares at the left bottom or right upper corner of each contact matrix represent the minimum interaction frequency for the reddest locus pair. The black arrow highlights one example region that keeps showing over-imputation artifacts by Higashi across all five cell types compared to the bulk Hi-C data as a gold standard. First row: The insulation scores [36] that trace the TAD boundaries are depicted on the contact matrices with gray lines corresponding to bulk Hi-C data and purple for BandNorm, blue for scVI-3D, green for Higashi, and yellow for scHiCluster. Second row: A/B compartments are detected using the eigenvector of the correlation map of bulk (upper right triangles) or aggregated (lower left triangles) Hi-C matrices, values of which are displayed above each correlation matrix

bins. Both strategies reduce the heterogeneity of the cells across cell types. Imputation across neighbor cells relies strongly on the accurate separation of the cells based on their cell types to form informative neighborhoods. Over-expanding the neighbor cells network and over-smoothing across neighbor bins can lead to homologous interaction patterns across different cell types. As revealed by the UMAP and t-SNE visualization of cell type clustering (Fig. 2b and Additional file 1: Fig. S11), none of the methods can isolate IMR90 cells, and most IMR90 cells are mixed with the HFF cell cluster. As a result, neighbors of IMR90 cells inevitably consist of cells from various cell types and are especially enriched in HFF cells. Furthermore, as discussed in Fig. 2b, Higashi failed to assign cells with relatively low sequencing depth and high sparsity to their correct cell type clusters, which in turn impacted the purity of cells in the neighboring network.

A use case of scHi-C data is the inference of similarity between different cell types based on their 3D chromatin interactions. We evaluated the hierarchical relationship of the aggregated scHi-C normalized or de-noised matrices across 14 cell types of *Lee2019* data set constructed based on the HiCRep similarity scores. Fig. 7d demonstrates that both BandNorm and scVI-3D can achieve a distinct separation of the excitatory neuron subtypes (L2/3, L4, L5, and L6), CGE-derived inhibitory subtypes (Ndnf and Vip), medial ganglionic eminence-derived inhibitory subtypes (Pvalb and Sst), oligodendrocyte related cell types (Astro, OPC, ODC) and non-neuronal cell types (MG, MP, Endo)

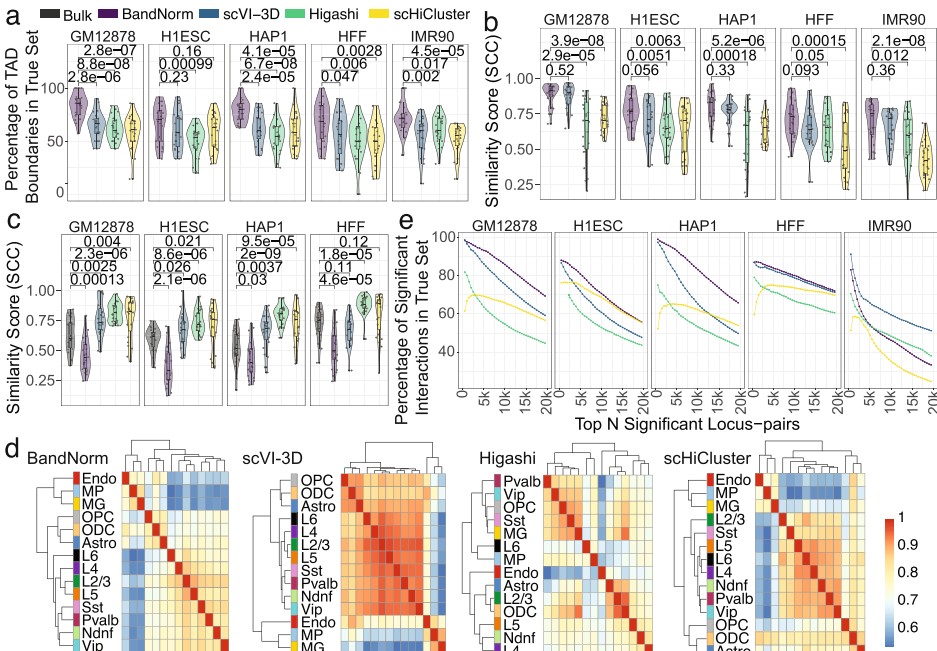

**Fig. 7** Evaluation of scHi-C normalization and de-noising methods for their impact on downstream analysis. **a** Percentages of TAD boundaries from *Kim2020* data set that are within 1Mb distance of the TAD boundaries of the same cell type bulk Hi-C data. The numbers above each pairwise comparison with BandNorm are the *p* values based on the two-sided t-test. **b** HiCRep [13] similarity scores between aggregated scHi-C matrices from the *Kim2020* data set and the corresponding bulk Hi-C matrices. The numbers above each pairwise comparison with BandNorm are the *p* values based on the two-sided t-test. **c** HiCRep [13] similarity scores between aggregated IMR90 matrices and aggregated GM12878, H1Esc, HAP1, and HFF from *Kim2020* data set. The numbers above each pairwise comparison with bulk are the *p* values based on the two-sided *t*-test. Sample sizes for **a–c** are n = 23 for each violin plot corresponding to chromosome 1-22 and chromosome X. **d** Hierarchical clustering of the aggregated scHi-C matrices of different cell types from BandNorm, scVI-3D, Higashi, and scHiCluster based on their pairwise HiCRep similarity scores, depicted in the heatmap matrices for *Lee2019* data set. **e** Percentage of top *N* (*N* = 5,000, 10,000, ...) significant interacting locus pairs that are in the gold standard set for each method on *Lee2019* data set. The gold standard set is defined as the top 50,000 significant locus pairs detected by Fit-Hi-C [37] from the cell-type-specific bulk Hi-C data

as suggested in Lee et al. 2019 [10]. Higashi and scHiCluster, however, fail to capture the cell type specificity, hence cannot recover the true cell type relationship.

We next investigated the performances of the de-noising methods at the locus pair level in terms of detecting significantly interacting locus pairs within a cell type and differentially interacting locus pairs across cell types. We first set a gold standard using the cell line specific bulk Hi-C data. Specifically, we identified the top 50,000 significant interacting locus pairs from bulk Hi-C with Fit-Hi-C [37] as the "true" significant interaction list. Then, we quantified the percentage of this list that can be recovered by the top interacting locus pairs from the aggregated scHi-C matrices of each method. BandNorm outperforms all the other methods by achieving the highest accuracy rate (e.g., for the top 5000 interacting locus pairs of GM12878, the accuracy rates vary as 93.34%, 86.32%, 63.66% and 69.26% for BandNorm, scVI-3D, Higashi, and scHiCluster, respectively), across all cell lines except for the IMR90 scHi-C data which has the smallest number of cells (Fig. 7e). scVI-3D maintains a highly significant interaction detection accuracy rate for the IMR90 cells owing to its zero-inflation model and successful imputation strategy (Fig. 7e). Using a series of FDR thresholds (FDR ≤ 0.001, 0.01, 0.05, 0.1), the number

of significant interacting locus pairs varies across cell lines and normalization methods, with BandNorm and scVI-3D having the highest percentage of the significant locus pairs from the aggregated samples in the bulk significance list (Additional file 1: Fig. S20). In order to evaluate performances for detection of differential TADs or locus pairs, we considered TADcompare [38] for differential TAD boundaries detection, diffHic [39] for differential interacting locus pairs detection, and CHESS [40] for differential interacting regions detection (Additional file 1: Fig. S21). Overall, aggregated scHi-C matrices from all four methods resulted in similar differential interaction detection.

### Single-cell gene associating domain (scGAD) score enables analysis at gene-level resolution

#### *High scGAD score is a salient feature of marker genes*

After normalization and de-noising of scHi-C data, we turned our attention to investigating single-cell 3D genome architecture at the gene level. Specifically, we adapted the gene associating domain (GAD) analysis on the bulk tagHi-C data [41] to scHi-C data. GAD scores, which quantify the ratio of the interaction frequency within the gene to the average interaction frequency of upstream and downstream regions of the same length, have been identified as general structures for highly expressed genes [41]. We utilized this gene domain concept on the *Tan2021* data set [11] to illustrate how single-cell gene associating domain (scGAD) [42] can reveal gene-level insights into 3D genome structure. The *Tan2021* data set harbors both diploid chromatin conformation capture (Dip-C) data and high-resolution multiple annealing and looping based amplification cycles for digital transcriptomics (MALBAC-DT) of post-natal brain development in mice and is particularly appealing for assessing whether insights from scGAD analysis can be corroborated with the single-cell transcriptome data. Our benchmarking of the scHi-C data normalization and de-noising methods for cell type separation revealed that BandNorm demonstrated consistently outstanding performances for data analyzed at adequate resolution. Notably, *Tan2021* has one of the highest genome coverages and sequencing depths compared to other scHi-C assays, and it enables binning at 100kb without encountering sparsity issues (Additional file 1: Fig. S23). As a result, low-dimensional projection of Dip-C contact matrices normalized by BandNorm achieves a level of cell-type separation (Fig. 8a and Additional file 1: Fig. S24A) which is validated by a similar cell-type separation from single-cell transcriptomics with a reanalysis of the MALBAC-DT as in Tan et al. 2021 [11] (Fig. 8b).

Next, we computed scGAD scores [42] and observed that scGAD scores are consistently higher for the cell-type specific marker genes detected by MALBAC-DT data (Fig. 8c). Empirical correlations between scGAD and scRNA-seq expression of matching cell types (Additional file 1: Figs. S25-S28) further confirmed that scGAD is a salient feature of highly expressed genes. We next leveraged the scGAD scores to derive cell-type-specific marker genes based on 3D genome architecture using the cell-type-specific marker genes detection procedure applied to MALBAC-DT data in Tan et al. 2021 [11]. We observed a markedly significant overlap between these two sets of marker genes (detected by single-cell transcriptomics and scGAD scores from Dip-C, respectively) (Fig. 8d and Additional file 1: Fig. S29). In addition, the gene ontology (GO) enrichment analysis of scGAD marker genes revealed processes congruent with the

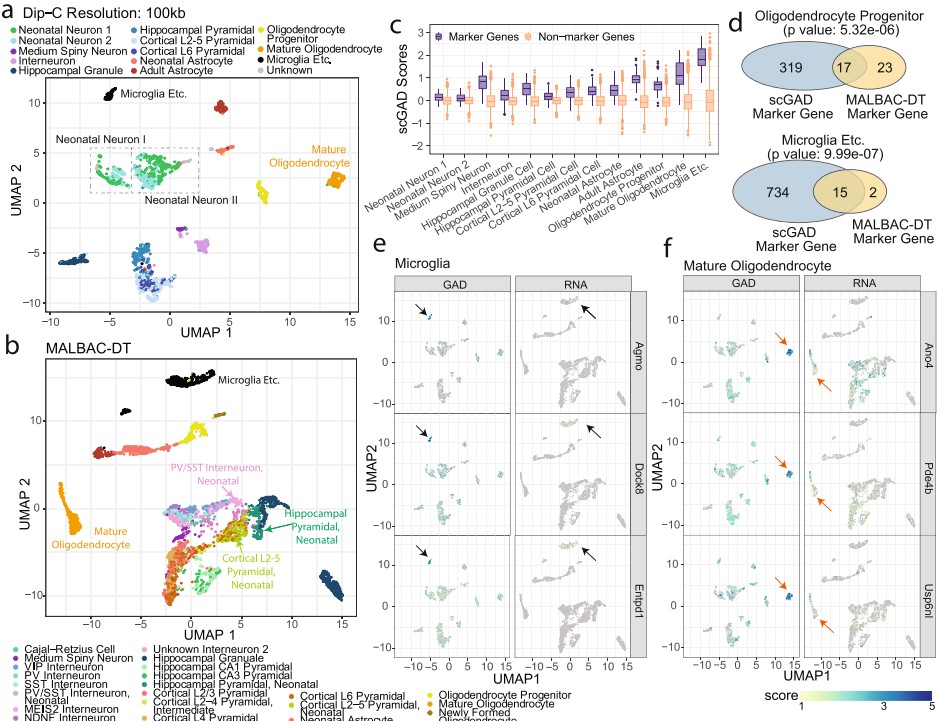

**Fig. 8** Single-cell gene associating domain analysis links single-cell 3D genome organization with single-cell transcriptome. **a** Single-cell 3D genome Dip-C data of postnatal mouse forebrain cell types from *Tan2021* data set [11]. UMAP visualization is generated after BandNorm normalization of single-cell contact matrices. **b** UMAP visualization of single-cell transcriptomics MALBAC-DT data of postnatal mouse forebrain cell types from *Tan2021* data set. **c** Comparison of single-cell gene associating domain (scGAD) scores between cell type-specific marker genes detected by the MALBAC-DT data and non-marker genes. **d** Overlap between cell-type-specific marker genes detected using scGAD scores and single-cell transcriptomics (MALBAC-DT) gene expression. *P* values are obtained by Fisher's exact test. **e-f** scGAD scores and gene expression values of marker genes that are specifically detected by the scGAD scores for Microglia (**e**) and Mature Oligodendrocyte (**f**) cells, highlighted by the pointing arrows

cell-type-specific functions (Additional file 1: Fig. S30), further validating scGAD quantification as a way of deriving cell-type-specific marker genes. For example, the top biological processes for marker genes of microglial cells are related to the immune system process, and immune response, consistent with the role of microglia cells as the active immune defense macrophage cells in the central nervous system [43]. Additionally, central nervous system neuron differentiation and transport are the leading biological processes for cortical L2-5 pyramidal cells. The major cellular component of scGAD marker genes for oligodendrocyte progenitor cells is the highly relevant synaptic vesicle membrane [44] and the top biological process is the chondroitin sulfate proteoglycan biosynthetic process which produces hallmark protein chondroitin sulfate proteoglycan 4 of oligodendrocyte progenitor cells [45, 46]. scGAD analysis also provides additional cell-type identification utility when the expressions of scGAD marker genes are generally low or not specific to the target cell type, as we illustrate in microglia and mature oligodendrocyte cells (Fig. 8e–f). Genes such as Agmo, Dock8, and Entpd1 are among the top 10 scGAD marker genes of microglia cells with adjusted *p* values below $10^{-30}$. However, Agmo, detected as one of the top cell surface candidate genes enriched for microglia

cells in a virus-induced neuroinflammation study [47], ranks below 700 using MALBAC-DT gene expression data, barely making it on the cell-type-specific transcriptome-based marker gene list. Moreover, Dock8 is a marker gene of mouse brain macrophages cells [48], and its microglial regulation activities have been extensively investigated [49]. Entpd1, also known as CD39, is a microglia signature gene [50] and has been shown to promote depression behavior in mice [51]. Both Dock8 and Entpd1 are slightly outside of the transcriptome-based marker gene list. Similarly, Ano4, Pde4b, and Usp6nl rank 1st, 2nd, and 32nd for the scGAD marker gene detection with adjusted $p$ values all below $10^{-32}$ albeit without mature oligodendrocyte specific gene expression according to the MALBAC-DT data. However, Allen Brain Atlas browser 10X-SMART-SEQ taxonomy data [52] confirms the enrichment of Ano4, Pde4b, and Usp6nl gene expression specific to post-natal Oligodendrocyte cells. Altogether, these underline the potential of scGAD analysis to reveal cell-type-specific marker genes based on 3D genome architecture data.

### scGAD scores of marker genes support novel clustering of neonatal neuronal cells based on 3D genome architecture

Analysis in Tan et al. 2021 [11] separated neonatal neuronal cells into two sub-clusters based on their 3D genome architecture as neonatal neuron 1 and 2 (Fig. 3A of Tan et al. 2021 [11]). This clustering is primarily driven by age as 78.78% of neonatal neuron 1 cells originate from day 1 and 68.8% of neonatal neuron 2 from day 7. After BandNorm normalization, both UMAP and t-SNE projections reveal a novel cluster separation among neonatal neuronal cells (labeled as neonatal neuron I and II in Fig. 8a and Additional file 1: Fig. S24A). To investigate the biological relevance of this sub-clustering, we first ruled out that tissue (cortex or hippocampus), age (day 1 or day 7), sex (female or male), and sequencing depth are the driving factors for this sub-cluster separation (Additional file 1: Fig. S31). Next, leveraging the ability of scGAD scores to accentuate marker genes, we asked whether any of the cell-type-specific transcriptome-based marker genes of the neonatal-related cell types (Additional file 1: Fig. S32 and Section 5) has differential scGAD scores between the two sub-clusters. This did not reveal any marker gene that is able to separate these two sub-clusters from Tan et al. 2021 [11] (neonatal neuron 1 vs. 2). In contrast, 50 of the transcriptome-based marker genes had significantly different scGAD scores between the two sub-clusters with the new cluster labeling on the BandNorm normalized data (neonatal neuron I vs. II, Fig. 9a). We further partitioned these 50 marker genes into two groups based on the direction of their differential scGAD scores for the two sub-clusters. Then, we investigated the single-cell expression of these genes in three major neonatal neuron-related cell types: PV/SST neonatal interneuron, hippocampal pyramidal neonatal, and cortical L2-5 pyramidal neonatal cells (Figs. 8b and 9b). As depicted by the black rectangles on the heatmaps, a cluster of marker genes with significantly higher scGAD scores in the neonatal neuron I sub-cluster show high expression among interneuron neonatal cells, suggesting that the neonatal neuron I sub-cluster of the 3D genome architecture might be enriched for interneuron neonatal cells. In contrast, marker genes with significantly higher scGAD scores in the Neonatal Neuron II sub-cluster exhibit an exclusive expression pattern for the interneuron neonatal cells, especially during day 1, suggesting that the neuron II sub-cluster of the 3D genome architecture might be depleted of interneuron neonatal cells. Furthermore, the expression patterns of the marker genes highlight that neonatal neuron I sub-cluster is likely to exclude the day 1 (P1) versions of the other two neonatal cell types

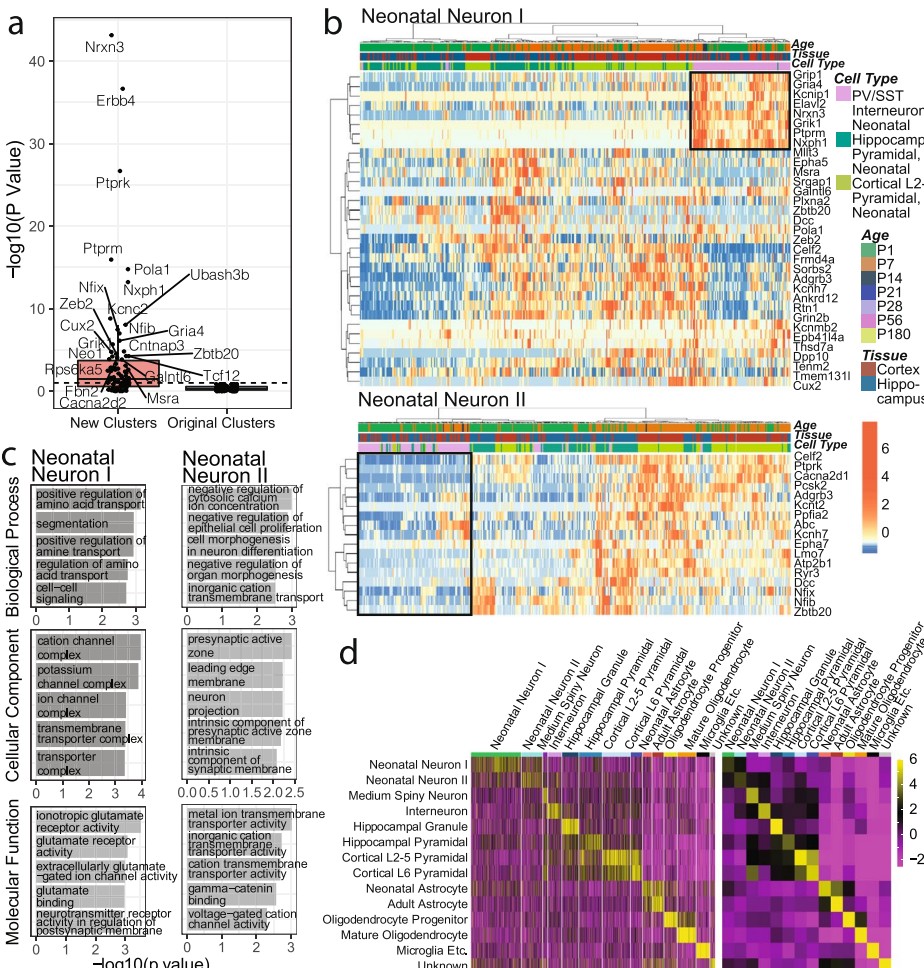

**Fig. 9** Detection of two novel neonatal neuronal sub-clusters after BandNorm normalization. **a** Transformed *p* values from the comparison of scGAD scores of marker genes across the two sub-clusters. "New clusters" and "Original clusters" depict the sub-cluster labels obtained in this paper and in Tan et al. 2021 [11], respectively. The marker genes are inferred by Tan et al. 2021 [11] from the neonatal single-cell transcriptomics data. *P* values are obtained by Wilcoxon rank-sum test. **b** Heatmaps of gene expression of the marker genes with differential scGAD scores between the newly identified neonatal neuron I and II sub-clusters among the PV/SST interneuron, hippocampal pyramidal, and cortical L2-5 pyramidal neonatal cells. Marker genes with significantly different scGAD scores among the two sub-clusters are partitioned into two groups. Top heatmap: genes with significantly higher scGAD scores in the newly identified neonatal neuron I sub-cluster compared to neonatal neuron II sub-cluster. Bottom heatmap: genes with significantly higher scGAD scores in the newly identified neonatal neuron II sub-cluster compared to neonatal neuron I sub-cluster. **c** Gene ontology analysis of marker genes with significantly different scGAD scores between Neonatal Neuron I and II. **d** scGAD scores of lists of cell-type-specific marker genes, detected by comparing scGAD scores across cell types among single cells (left panel) and averaged for each cell type (right panel). scGAD score values are standardized into *z*-scores

(hippocampal pyramidal and cortical L2-5 pyramidal neonatal) and harbor more mature (P7) versions of these cell types (Fig. 9b). GO enrichment analysis of the two groups of marker genes for neonatal neuron I and II also exhibit different biological processes, components, and functions, further supporting the feasibility of novel neonatal neuron sub-clustering (Fig. 9c). Collectively, this analysis highlights the utility of scGAD scores in interpreting new cell clusters revealed by clustering of the 3D genome architecture (Additional file 1: Fig. S24C) and

cell-type identification at individual cell resolution (Fig. 9d - left panel) or using the averaged scGAD scores across cells of the same cell type (Fig. 9d—right panel).

### Run time and memory requirements

Finally, we quantified the run time and memory requirements of each method on the *Lee2019* data set with large numbers of cells and high sequencing depths. Higashi experiments using GPU version and scVI-3D were carried out on a machine with 18 cores 2 sockets Intel(R) Xeon(R) Gold 6254 CPU addressing 562GB RAM and one NVIDIA GeForce RTX 2080 Ti GPU addressing 11GB RAM. The rest of the methods, including Higashi experiments only using CPU, were tested on machines with 14 cores 2 sockets Intel(R) Xeon(R) CPU (E5-2680 v4) addressing 256GB RAM. The run times of the methods vary dramatically from less than 15 min to several hours or even days using multiple cores for parallel running with one core per chromosome (Table 2).

### Discussion

The profiling of single-cell 3D genome organization is poised to generate new types of investigations of transcriptional regulation at the single-cell level. A critical analytical task of these investigations is de-noising and normalization of scHi-C data to infer clusters of cells representing cell types, stages, and conditions. We developed BandNorm and scVI-3D at the two opposite ends of the structured modeling of scHi-C data. Our evaluations of data sets with known cell types and varying data characteristics indicate that BandNorm, which corrects for genomic distance bias and library size, performs as well and even better than elaborate modeling approaches on these data sets. When coupled with Harmony, BandNorm is also robust to batch effects in cell type separation. In comparison, scVI-3D can account for genomic distance, library size, sparsity, and batch effects and impute sparse contacts to provide advantages in addressing high sparsity or rare cell types for downstream analysis compared to BandNorm. Collectively, our computational investigations suggested that BandNorm would be effective for the initial exploration of the scHi-C data with normalization and cell-type identification. If and when rare cell types are identified, detailed modeling with scVI-3D can enable imputation and better inference for downstream analysis, especially for high sparsity settings. We also expect that the framework of scVI-3D could foster further model-based approaches for the analysis and integration of scHi-C data with other data modalities.

To enable exploration of scHi-C data at the gene level, we adapted gene-associating domain analysis to single-cell level as scGAD scores and leveraged these to infer highly expressing and/or marker genes of cell types. Our current implementation of scGAD scores identified marker genes from 3D genome organization clustering, largely agreeing with cell-type-specific marker genes inferred from single-cell transcriptome analysis. We further illustrated how scGAD scores could elucidate cell types in a novel sub-clustering of neonatal neuronal cells. scGAD scores are calculated for genes of length at least 100kb owing to the low resolution of chromatin conformation capture data. As the resolution of single-cell 3D genome organization data improves, we expect that the length constraint for scGAD scores can be removed by careful design of background distributions to take into account gene densities.

We note that the current version of scVI-3D does not consider spatial dependence among adjacent locus pairs, i.e., $(i, j)$ with $(i - 1, j - 1)$ and $(i + 1, j + 1)$ in the band matrices. While

high-resolution contact maps, e.g., locus size $\leq$ 10Kb from bulk Hi-C data exhibit local spatial dependency between interacting loci [53], the spatial independence assumption is well justified for low resolution (200 Kb–1 Mb) scHi-C data (Additional file 1: Fig. S33).

## Conclusions

In conclusion, we have presented BandNorm as an initial tool for exploring and scVI-3D for more structured denoising of scHi-C data. Compared to existing approaches, both methods performed as well or markedly better than existing approaches with specific time and memory efficiency advantages for BandNorm and the ability to adapt to data sparsity and cell type rarity for scVI-3D. Our re-analysis of the single-cell 3D genome data from post-natal brain development in mice [11] together with scRNA-seq data from the same system further illustrated the potential of integrating scHi-C data with other single-cell data modalities and highlighted an exciting emerging direction [42].

## Methods

### Band transformation of scHi-C data

Band transformation of scHi-C contact matrices forms the basis for our normalization and modeling approaches. To explicitly capture the genomic distance effect, the upper triangular of the symmetric contact matrix for each cell is first stratified into diagonal bands, each representing the genomic distance between the interacting loci. Then, bands from the same genomic distance are combined into a band matrix across cells (Fig. 1a). Specifically, we denote the set of bands by $\mathcal{V} = \{0, 1, \cdots, D-1\}$, where $D$ denotes the number of loci in the contact matrix (i.e., the number of rows) and $v = 0$ represents the diagonal band, $\mathcal{A}(v) = \{(i,j), i,j = 1, \cdots, D : j \geq i \quad \& \quad j - i = v\}$ indices of the locus pairs in the band $v$, and $m_v$ the total number of locus pairs in the band $v$. We only consider the off-diagonal interactions for the modeling and downstream analysis, hence $v \geq 1$. Therefore, each element in the band matrix at genomic distance $v$ is denoted as $Y_r^{cv}$ representing the raw interaction frequency, i.e., quantification of how strongly two loci interact, between genomic loci $i$ and $j$ where $r \in \mathcal{A}(v)$ in cell $c$ ($c = 1, \cdots, N$). $\mathbf{Y}^{cv}$ denotes a vector of interaction frequencies of locus pairs for cell $c$ at band $v$.

### BandNorm: a fast baseline band normalization approach for scHi-C data

Bandnorm provides computationally fast normalization of scHi-C data and operates by first removing genomic distance bias within a cell and sequencing depth normalizing between cells followed by adding back a common band-dependent contact decay estimate for the contact matrices across cells. More specifically, the BandNorm normalized interaction frequencies are obtained by

$$C_r^{cv} = \frac{Y_r^{cv}}{L^{cv}/m_v}(\alpha(v)/m_v) = \frac{Y_r^{cv}}{L^{cv}}\alpha(v), \quad \forall r \in \mathcal{A}(v), \tag{1}$$

where $L^{cv}$ denotes the total interaction frequency of cell $c$ at the $v$-th band, $L^{cv} = \sum_r^{m_v} Y_r^{cv}$, and $\alpha(v)$ is the average interaction frequency of band $v$ across cells and is defined as $\sum_{c=1}^{N} L^{cv}/N$.

### scVI-3D: a deep generative model for scHi-C data

scVI-3D models the interaction frequencies of locus pairs in each cell as a sample from a zero-inflated negative binomial distribution while accounting for library size and batch effect for each band matrix. The two key components of this model are a non-linear latent factor model to obtain low-dimensional representations $\mathbf{z}_{cv}$ of cell $c$ across band matrices $\mathbf{Y}^{cv}$ and a hierarchical generative model for $Pr(\mathbf{Y}^{cv} \mid \mathbf{z}_{cv})$. Low-dimensional latent variable $\mathbf{z}_{cv}$ enables nonlinear dimension reduction for characterizing differences among cells in band $v$. In the generative process, each interaction frequency $Y_r^{cv}$ is drawn independently conditional on $\mathbf{z}_{cv}$ through the following process, assuming that $\mathbf{z}_{cv} \sim \text{Normal}(0, I_K)$. In order to account for the sparsity of the scHi-C data, a zero inflation variable $T_r^{cv}$ is defined while setting $Y_r^{cv} = 0$ if $T_r^{cv} = 1$ and $Y_r^{cv} = N_r^{cv}$ otherwise. Here, $N_r^{cv}$ denotes the observed interaction frequency in the absence of a dropout, and $T_r^{cv} \sim \text{Bernoulli}(\pi_r^{cv})$, where $\pi_r^{cv} = \omega_r^v(\mathbf{z}_{cv}, \mathbf{s}_c)$. $\mathbf{s}_c$ is the batch information for cell $c$ and $\omega(.)$ is a neural network that encodes whether a particular locus pair has dropped out due to the technical artifacts [29] and maps the latent space $\mathbf{z}_{cv}$ back to the full dimension of all locus pairs in band $v$. Additionally, $N_r^{cv} \sim \text{Poisson}(l_c b_v \lambda_r^{cv})$ where $\lambda_r^{cv} \sim \text{Gamma}(\mu_r^{cv}, \delta^v)$, $l_c$ denotes the latent library size factor for cell $c$, and the band size factor $b_v$ modulates the impact of the size factor on the true interaction frequencies for band $v$. The band size factor, $b_v$, is motivated by the genomic distance effect in which interaction frequencies between locus pairs vary systematically by the distance between the loci. The band effect has been observed to vary markedly between two bulk Hi-C experiments [30]. We inquired whether this effect varied depending on the observed library size of the individual cells by leveraging one of our case study data sets (*Lee2019*) and formally tested for an interaction between library size and band. Specifically, we considered a mixed linear model for cell-specific mean band interaction frequencies as a function of cell type, observed library size, and band indices as

$$
\begin{aligned}
\text{Normalized mean band IFs} \\
\sim \text{Cell Type} + \text{Library Size} + \text{Band} \\
+ \text{Library Size} \times \text{Band} \\
+ (1|\text{Cell Number}),
\end{aligned}
\tag{2}
$$

where interaction frequencies are normalized to per million within each cell and log transformed, and the model term `(1|Cell Number)` denotes the random effects of the cells to accommodate potential correlations between measurements from the same cell. This analysis revealed a significant interaction between library size and band ($p$ value $<< 10e-6$), and the model with the interaction term fitted better than the smaller submodels based on the Bayesian information criterion (BIC [54]). This observation enables a more flexible parametrization of scaling factor that merges library ($l_c$) and band size ($b_v$) factors into cell type specific band size factors as $d_{cv}$. Moreover, we let $d_{cv} \sim \text{Log Normal}(\mu_d^v, (\sigma_d^v)^2)$ parametrize the prior for this scaling factor in the

generative model. Next, the mean parameter $\mu_r^{cv}$ is modeled as a nonlinear function of $\mathbf{z}_{cv}$ as $\mu_r^{cv} = \eta_r^v(\mathbf{z}_{cv})$, where $\eta(.)$ is a neural network that maps the latent space to the full dimension of the locus pairs.

Computationally, we extended the existing variational inference tools, scvi-tools (single-cell variational inference tools [18]), to fit on each chromosome and each band matrix separately. The estimated latent components $\mathbf{z}_{cv}$ of each cell and band were concatenated for final low-dimensional projection by UMAP or t-SNE and downstream analysis. Prior to the modeling, we filtered each band matrix to exclude the cells with no interaction across all the locus pairs within each band. Consequently, the latent embeddings were missing for the no interaction cells. We impute the missing embeddings by 0. The scVI-3D software offers an option to tune the dimension of latent space; however, we suggest starting with the default value of 100 as computational experiments illustrate advantages over the commonly used 10–50 dimensions in scRNA-seq or scATAC-seq analysis (Additional file 1: Fig. S4). Furthermore, we also offer optional usage of the band pooling strategy to combine the farther off-diagonals for gathering sufficient locus pairs and interaction frequencies for the model fitting to perform more robust dimension reduction and de-noising. The default pooling strategy leverages the progressive pooling concept from multiHiCcompare [24], where the diagonal is modeled on its own. The second and third off-diagonal of the band matrices are concatenated, followed by combining the fourth, fifth, and sixth diagonals. Then, the 7th to 10th diagonals are concatenated, and so on. The last group pools all the remaining off-diagonal band matrices, and all the bands concatenated together share the same training parameters. We also explored four additional pooling strategies (Additional file 1: Fig. S3). Strategy 2 models the diagonal and first nine off-diagonal independently and then pools the 11th and 12th bands, 13th to 15th, 16th to 19th, and so on. Strategy 3 merges the first five diagonals, then the next ten off-diagonals, followed by the next 20. Strategy 4 merges every ten bands, and strategy 5 models the first ten bands individually and then starts merging every ten bands subsequently. Computational investigation illustrates that the default strategy generally demonstrates the leading performances in better separating the cell types while exhibiting minor batch effects (Additional file 1: Fig. S3). We also considered applying a similar pooling strategy for BandNorm. However, the genomic distance decay profiles that are critical for BandNorm normalization became less precise (Additional file 1: Fig. S34), and we observed deteriorating cell type separation performance (Additional file 1: Fig. S3). Therefore, progressive pooling is not incorporated into BandNorm.

### scHi-C data analysis methods compared in the benchmarking experiments

In our evaluations of unsupervised clustering of the cells based on scHi-C data, we considered two classes of methods, including baseline methods for library size and genomic distance effect normalization and more structured modeling approaches. In the former category, in addition to BandNorm, we devised and evaluated CellScale, which uses a single scaling factor across all the locus pairs within a cell and scales the library sizes of each cell to a common size (10,000 in our applications). We also

considered the first normalization step of BandNorm separately as BandScale, where interaction frequencies of each band within a cell are divided by the cell's band means. BandScale uses band-specific size factors rather than a global size factor within a cell and has been used previously to eliminate library size bias at each genomic distance [10, 23, 24]. After each of CellScale, BandScale, and BandNorm normalizations, single-cell contact matrices are vectorized into the cell by locus pair matrices and used to generate low-dimensional embeddings. To incorporate the matrix structure of the data, we utilized a convolutional neural network (CNN) approach, which has been previously leveraged for enhancing the resolution of the bulk-cell Hi-C matrix [31], on contact matrices from CellScale to learn the lower-dimensional representation of the contact matrices.

   Among the more structured modeling approaches, in addition to scVI-3D, we also considered the state-of-the-art scHi-C data processing methods scHiCluster, scHiC Topics, and Higashi. scHiCluster starts with neighborhood smoothing and random walk imputation to reduce the contact matrix sparsity. For dimension reduction, the contact matrix of each chromosome is vectorized into a cell by a locus pair matrix, and after concatenating across chromosomes, truncated singular value decomposition (SVD) is leveraged for dimension reduction, followed by clustering and visualization. Notably, scHiCluster requires the most stringent cell filtering, where only cells with total off-diagonal interaction frequencies > 5000 are kept. It further enforces less sparsity by discarding cells with less than $x$ non-zero locus pairs, where $x$ is the number of $x$ Mb loci on each chromosome, separately for the contact matrices of each chromosome. scHiC Topics focuses more on the short to mid-range interactions by only considering the intra-chromosomal locus pairs within 10Mb genomics distance. This aims to balance the data sparsity and reduce model complexity. Contact matrices of the cells are first vectorized to construct the cells by locus pairs matrix, which is further binarized and input into a topic modeling framework. Cell type clustering is implemented on the cell by topics matrix where "topics" are proxies for cell types. Higashi trains a hypergraph neural network and enables neighboring cells in the hypergraph to share information for capturing interaction patterns. The resulting embeddings are then used for learning cell types. Table 2 summarizes these eight methods in terms of their pre-processing and treatment of various sources of biases.

### Benchmark and analysis data sets

We considered four existing studies with varying data characteristics and known cell types to benchmark the scHi-C low-dimensional embedding approaches. These four datasets are scHi-C measurements from human cell lines (*Ramani2017* [8] and *Kim2020* [15]), human brain prefrontal cortex cells (*Lee2019* [10]), and mESC cells (*Li2019* [9]). We utilized one additional cortical and hippocampal mouse study *Tan2021* [11] to explore single-cell 3D genome information with scGAD scores and to validate with scRNA-seq data.

   *Ramani2017* has four human cell lines, HeLa S3, HAP1, K562, and GM12878. These cell lines are distributed over five sequencing libraries labeled as ml1, ml2, ml3, pl1, pl2, where pairs ml1 and ml2, and pl1 and pl2 are sequencing experiments with the same library preparations, respectively, and hence present different batches. We downloaded

the *Ramani2017* data from GEO [55] with data accession GSE84920, and followed the instructions of literature [8] to filter out low count reads. Data was transformed into the sparse matrix format at 1Mb by scHiCTools [56].

*Lee2019* generated scHi-C data from 14 human brain prefrontal cortex cell types, Astro, Endo, L2/3, L4, L5, L6, MG, MP, Ndnf, ODC, OPC, Pvalb, Sst, Vip, originating from two donors with ages of 21 and 29 years and in a total of five sequencing libraries. Data were downloaded from https://salkinstitute.app.box.com/s/fp63a 4j36m5k255dhje3zcj5kfuzkyj1/folder/82405563291. This dataset has a relatively large number of sequencing reads and a high average interaction frequency per cell. Furthermore, since all the cells are prefrontal cortex cells, they are expected to exhibit less heterogeneity compared to other datasets.

*Li2019* dataset harbors mESC cells cultured in serum and leukemia inhibitory factor (LIF) condition (serum mESCs: serum 1 and serum 2) and mESCs cultured in LIF with GSK3 and MEK inhibitors (2i) condition. This data is valuable in benchmarking the performances of the methods when the number of cells is small. We downloaded the *Li2019* data from GEO under the accession number GSE119171 and converted it into sparse contact matrices by Juicer [57].

*Kim2020* dataset contains scHi-C data from five human cell lines, GM12878, H1Esc, HAP1, HFF, and IMR90, with nine sequencing libraries. While this data has the largest number of cells, the average off-diagonal interaction frequency per cell is the smallest. Notably, the numbers of cells vary dramatically across cell types, with GM12878, H1Esc, and HAP1 having more than 2k cells and IMR90 with less than 100 cells (Additional file 1: Fig. S1). We downloaded the data from https://noble.gs.washington.edu/proj/ schic-topic-model/.

All the scHi-C data on chromosomes 1-22 for human cell lines and 1-19 for mouse cell lines and chromosome X were binned at 1Mb resolution (default resolution of all the analyses unless otherwise specified) to generate a set of loci, and extremely sparse cells were removed if the number of non-zero locus pairs was less than $x/6$ for the contact matrix of each chromosome where $x$ is the chromosome size in Mb (Table 1). We also added one additional investigation of *Lee2019* at 100kb for inspection of method performance at high resolution and high sparsity scenario. At 100kb, cells that have more than 99.9% of sparsity rate, i.e., more than 99.9% interacting bins on the within-chromosome contact matrix are zero, are excluded. We discarded the scHiCluster cell filtering requirement (Table 2) since it led to the removal of as high as 88.4% of the cells in the *Kim2020* dataset. The valid numbers of cells per cell type in all four data sets are summarized (Additional file 1: Fig. S1). As part of pre-processing, all the locus pairs along the diagonal of the contact matrices were excluded from the analysis. We observed distinct distributions of interaction frequencies among the diagonal and off-diagonal locus pairs (Additional file 1: Fig. S35) across the datasets. In the benchmarking experiments, locus pairs at all genomic distances (excluding the diagonals) were utilized to avoid the exclusion of large percentages of interactions (Additional file 1: Fig. S36), except for scHiC Topics, which only focus on locus pairs within 10Mb.

We used existing bulk Hi-C datasets of specific cell types as the gold standard for comparing different methods. Specifically, GM12878 combined in situ and IMR90 combined in situ data were downloaded from GEO under accession GSE63525 [53]. For differential

interaction detection, two deeply sequenced biological replicates (replicates HIC019 and HIC020 for GM12878 and HIC051 and HIC056 for IMR90) were utilized. One combined replicate of HAP1 was obtained from GEO with the accession number GSE74072 [58] and another replicate utilized the wild-type condition data from GEO with accession number GSE95015 [59]. H1ESC (accession 4DNESRJ8KV4Q) and HFF (accession 4DNES2R6PUEK) data were obtained from the 4D Nucleome portal [60].

*Tan2021* dataset has both the single-cell 3D genome data profiled by diploid chromatin conformation capture (Dip-C) and single-cell transcriptomics data using high-resolution multiple annealing and looping-based amplification cycles for digital transcriptomics (MALBAC-DT) on the mouse forebrain cortex and hippocampus tissue during the first post-natal month. Processed Dip-C and MALBAC-DT data are available at GEO [55] under accession GSE162511. Tan et al. [11] report 13 cell types from the Dip-C data and 26 from MALBAC-DT (Fig. 8a, b). The mouse reference genome is GRCm38 and gene annotations were downloaded from the GENCODE (https://www.gencodegenes.org/mouse/) M19 release. We binned the genome at 100kb resolution and normalized the contact matrices by BandNorm before visualization and downstream analysis.

## Evaluation metrics

We considered both K-means clustering with and without low-dimensional projections with t-SNE and UMAP, and Louvain graph clustering [32] to identify cell types. The cluster number of K-means is the number of cell types with true labels (e.g., $k = 14$ for *Lee2019*). The clustering resolution of Louvain graph clustering is tuned with a binary search so that the final cluster number is close to the true number of cell types. Application without t-SNE and UMAP involved applying K-means and Louvain graph clustering on the vectorized cell by locus pairs matrices (for CellScale, BandScale, and BandNorm) or the method latent component embeddings. Applications with t-SNE and UMAP low-dimensional projections first leveraged PCA to reduce the dimensions of the vectorized cell by locus pairs matrices or method latent component embeddings to the top 50 principal components before applying t-SNE and UMAP to generate low-dimensional embeddings. We quantified the resulting clustering performances with adjusted rand index (ARI) [61], which measures the similarity between two data clusterings, i.e., true underlying cell types and the estimated clustering, adjusted for chance similarity. We also evaluated average silhouette scores [62] to measure the separation between known cell types in the t-SNE and UMAP visualizations. Collectively, these led to six evaluation settings (Fig. 1c).

## Implementation details

### scHiCluster

We utilized the scHiCluster open-source software published at https://github.com/zhoujt1994/scHiCluster with the release version in December 2021. The software commit ID is 9c58381cb8d15dceafe410cd4b44cf3f376a475f. All the parameters are set to the default suggested by scHiCluster and followed the instructions to obtain the latent embeddings for all the chromosomes by the truncated singular value decomposition (SVD) extraction.

### scHiC Topic

Following the pre-filtering guidelines from scHiC Topics [15] (https://github.com/khj30 17/schic-topic-model), only locus pairs within 10Mb of each other were utilized for cell type separation. To determine the optimal topic number based on the silhouette score as suggested in the method paper [15], we set the lowest topic number to be 10 and the highest to be 90 with increments of 5. The final optimal number of topics selected for *Ramani2017, Li2019* and *Kim2020* are 35, and 60 for *Lee2019*, respectively.

### Higashi

We utilized the Higashi software of release version in October 2021 with the commit ID 19ec13f7da265f8d0780e3f6106b380056661991. All the parameters were set to defaults suggested by the Higashi pipeline (https://github.com/ma-compbio/Higashi). We benchmarked the performance of Higashi using GPU and only using CPU, therefore the `cpu_num` was set to 23 and `gpu_num` was set to 1 or 0 accordingly.

### CellScale+CNN

The CNN was implemented with some modifications to the CNN Variational Autoencoder module of Pytorch (https://github.com/sksq96/pytorch-vae). Both the encoder and decoder are symmetric and contain two layers. The latent vector $z$ was set to have 20 dimensions and `kernel_size` parameter to 4. Furthermore, `bias` was set to False and `stride` was set to 2 for all the layers. The minimal iteration parameter was set to 90 with a batch size of 10. The learning rate for the Adam algorithm was set to 0.001.

### scVI-3D

The implementation of scVI-3D on each band was built on the scVI-tools [18] (https://github.com/YosefLab/scvi-tools), where we set the latent variable dimension to 100. We filtered cells that have no interaction for all the locus pairs in each band matrix. When concatenating the latent embeddings across band matrices, the latent value for the missing cells was filled with zero.

### SVA

We first divided the obtained matrix by its minimum value to get a non-negative matrix and then use the *ComBat_seq* function in R package *sva* to do batch removal.

### Seurat

We constructed the *ScaleData* and utilized *vars.to.regress* function in R package *Seurat* to regress out the batch effect.

### Harmony

Use *HarmonyMatrix* function in R package *harmony* with *do_pca = FALSE* to remove the batch of the matrix. Harmony batch correction is utilized for all the scaling methods and more structured modeling approaches that do not explicitly model the batch

factor in their algorithms, namely, CellScale, BandScale, BandNorm, CellScale+CNN, and scHiC Topics.

### Integration local inverse Simpson's Index (iLISI) score

The iLISI [25] is employed with the "compute_lisi" function of the R package *LISI* to evaluate the batch effects. A high density of cells around one indicates that the neighborhoods of these cells have a single batch representation; hence, it signifies batch effects. In contrast, larger iLISI scores indicate that batches are well mixed in the cells' neighborhood, signaling a low impact of the batch effect.

### diffHic

Differential chromatin interaction detection requires at least two replicate per condition. For differential detection between two aggregated scHi-C cell types, we randomly partition the cells with the same cell label into two groups, with each forming a pseudo-replicate. Differential detection is implemented using R package *diffHic*.

### Evaluation metrics

K-means replied on the `kmeans` function of `stats` R package using the default `Hartigan-Wong` algorithm. `nstart` was set to 300 and `iter.max` to 1000. Louvain graph clustering was carried out based on `FindNeighbors` and `FindClusters` functions in `Seurat` R package. Silhouette coefficients were obtained using `silhouette` function from `cluster` R package.

### De-noising performances with aggregated scHi-C data using cell labels from unsupervised clustering

We repeated the assessment of the impact of the normalization and de-noising methods on the downstream analysis using the cell type labels inferred from clustering in aggregating the scHi-C contact matrices. This ensured a more unbiased assessment of the overall effect of the analysis methods without relying on true cell type labels (Additional file 1: Fig. S37). K-means clustering of the UMAP embeddings of the *Kim2020* dataset resulted in four clusters which we labeled as GM12878, H1ESC, HAP1, and HFF cell type (Additional file 1: Fig. S37a). Concordant with the results that relied on true cell labels, detection of A/B compartments and topologically associating domains (TADs), contact matrix similarity, and detection of interacting locus pairs yield the advantages of BandNorm followed by scVI-3D (Additional file 1: Fig. S37b-e). Similarly, comparison with respect to the differentially interacting locus pairs resulted in similar performances across the methods (Additional file 1: Fig. S37f-g).

### Single-cell gene associating domain (scGAD) analysis
### scGAD score definition

Inspired by the GAD analysis of bulk Hi-C data [41], single-cell gene associating domain (scGAD) adjusts the interaction frequencies within the gene region for gene-level sequencing depth and other potential genomic biases implicitly by a standardization procedure [42]. The scGAD score is calculated based on the individual cell contact

matrix after BandNorm normalization. Only protein-coding genes longer than 100kb are considered for scGAD scoring.

### Marker gene detection with scGAD

To detect the genes that have significantly high scGAD scores (i.e., marker genes with respect to scGAD scores), we first determined a threshold above which scGAD scores can be considered high using the same procedure as Zhang et al. 2020 [41]. Genes that are longer than 100kb are first ranked by the average scGAD scores across all the cells. Both the average scGAD scores and the gene ranks are scaled to [0, 1] to create a curve of scGAD scores as a function of scaled gene ranks. The threshold for a high scGAD score is then defined by deriving a tangent line to this curve with a slope of 1. The corresponding scGAD score before scaling to [0, 1] is set as the threshold to represent high scGAD scores (Additional file 1: Fig. S24B), resulting in a threshold of 1.638 for the Dip-C data. Then, for each cell type, we tested whether the mean scGAD score of genes exceeded this threshold using cell level scGAD scores and employing the one-sample Wilcoxon signed rank test. The resulting $p$-values were adjusted for multiplicity across genes within each cell type with the Benjamini-Hochberg procedure [63].

### scGAD cell-type specific marker gene detection

To recognize the marker genes that are specific to one cell type using scGAD score, we first normalize the scGAD scores using "NormalizeData" and "ScaleData" functions from Seurat R package. The detection is carried out using "FindMarkers" function from Seurat R package with the following parameters "only.pos = TRUE, min.pct = 0.1, logfc. threshold = 0.25," which are consistent with the single-cell transcriptomics cell-type marker gene detection in Tan et al. [11].

### scGAD differential marker gene detection between two neonatal neuron sub-clusters

Single-cell transcriptomics marker genes of five categories provided by *Tan2021* [11] are considered for the differential detection between two neonatal neuron sub-clusters of original or new cell type labeling using scGAD scores and tested by the Wilcoxon signed-rank test. These five categories are P1-7, neuron, PV/SST interneuron neonatal, hippocampal pyramidal neonatal, and cortical L2-5 pyramidal neonatal (Additional file 1: Fig. S32).

### GO analysis

Gene ontology analysis is performed by "goseq" function from goseq R package with reference data from "org.Mm.eg.db" and "TxDb.Mmusculus.UCSC.mm10.ensGene" R packages.

### Gene overlap analysis

The significance of the overlap between marker genes detected using single-cell transcriptomics data (MALBAC-DT) and those detected using scGAD scores are assessed by the Fisher's exact test.

## Supplementary Information

---

Additional file 1. Supplementary figures [81–83].

Additional file 2. Review history.

---

### Acknowledgements

We thank Dr. Peigen Zhou, Dr. Fan Chen, and Ms. Shan Lu from the University of Wisconsin - Madison, and Dr. Mehmet F Keleş from Johns Hopkins University for insightful discussions. We also thank the authors of Higashi, Dr. Jian Ma and Dr. Ruochi Zhang from Carnegie Mellon University, for sharing their analysis and useful discussions.

### Peer review information

### Review history

The review history is available as Additional file 2.

### Authors' contributions

SK conceived the project. YZ, SS, and SK designed the research and developed the methods. All the authors contributed to the preparation of the manuscript and approved the final manuscript.

### Funding

This work was supported by NIH grants HG003747 and HG011371 to SK.

### Availability of data and materials

R package, BandNorm, together with the curated scHi-C data sets are available at https://github.com/keleslab/BandNorm [64]. scVI-3D pipeline is implemented in Python in a way that allows parallelization in high-performance computing environments with source codes and instructions available at https://github.com/keleslab/scVI-3D [65]. The scripts used for running BandNorm, scVI-3D, and other software discussed in the manuscript and the analysis codes are available at https://github.com/keleslab/BandNorm_and_scVI-3D_manuscript [66]. All the GitHub repositories of BandNorm, scVI-3D, and the analysis codes for the manuscript are released under the GNU General Public License v3.0. The source codes are also available for download on Zenodo at https://doi.org/10.5281/zenodo.7076993 [67] for BandNorm, https://doi.org/10.5281/zenodo.7076777 [68] for scVI-3D, and https://doi.org/10.5281/zenodo.7084396 [69] for the analysis codes. This work used previously published data, which are explained in detail in the "Benchmark datasets" section. The data are available from several public repositories and are summarized below.

**Single-cell Hi-C data:**
- *Ramani2017* [70]: GEO accession number GSE84920.
- *Lee2019* [71]: GEO accession number GSE130711.
- *Li2019* [72]: GEO accession number GSE119171.
- *Kim2020* [73]: https://noble.gs.washington.edu/proj/schic-topic-model/.
- *Tan2021* [74]: GEO accession number GSE162511.

**Bulk-cell Hi-C data:**
- GM12878 [75]: GEO accession number GSE63525.
- IMR90 [76]: GEO accession number GSE63525.
- HAP1 [77, 78]: GEO accession number GSE74072 and GSE95015.
- H1ESC [79, 80]: 4D Nucleome portal [60] accession number 4DNESRJ8KV4Q and 4DNES2R6PUEK.

## Declarations

### Ethics approval and consent to participate

Not applicable.

### Consent for publication

Not applicable.

### Competing interests

The authors declare that they have no competing interests.

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

## 
