## [Additional file 2. Review history. · Genome Biology]

Review History

First round of review

Reviewer 1

Were you able to assess all statistics in the manuscript, including the appropriateness of statistical tests used? Yes: Statistics is appropriate.

Were you able to directly test the methods? No.

Comments to author:

The manuscript "Normalization and De-noising of Single-cell Hi-C Data with BandNorm and 3DVI" presents two single-cell Hi-C data analysis methods. BandNorm is a normalization approach, and 3DVI is a deep generative modeling framework utilizing Poisson and Negative Binomial distributions to model scHi-C counts accounting for library size and batch effect for each band matrix to minimize technical biases and denoise scHi-C data. It also uses gene-centric analysis of scHi-C data by adapting the concept of single-cell gene-body associating domain (scGAD) scores. The methods were benchmarked on four public scHi-C datasets using the band-specific scaling (BandScale), the global scaling (CellScale), BandScale+CNN, scHiCluster, scHiC Topics, Higashi. Cell type separation, addressing batch effect, the effect of sequencing depth/sparsity, several metrics were evaluated. Investigation of the impact of normalization and denoising on downstream analysis, concordance in A/B compartments, TAD boundaries (insulation score), agreement between different cell types, agreement between differential TAD boundaries and differential interactions. Single-cell gene-body associating domain (scGAD) score analysis allows for gene-level analysis and supports novel clustering enabling by the BandNorm normalization. The manuscript contains lots of information supported by figures and methods.

Bands' definition is unclear. Each subsequent diagonal band will be shorter than the previous and will be just a scalar at D. Shorter vectors will be more affected by outliers, especially pronounced in farther off-diagonal vectors. A similar approach, multiHiCcompare, progressively combines bands farther off-diagonal, which they dubbed 'progressive pooling'. How the problem of outliers, especially at larger distances, has been addressed/investigated, and whether a 'progressive pooling' strategy is warranted?

p.3 l.48 - when citing the BNBC method, two references seem missing. The BNBC paper itself refers to the HiCcompare paper [PMID: 30064362] which introduces distance-focused normalization. And, the aforementioned multiHiCcompare paper explicitly used bands for normalization and differential analysis.

Equation 1 needs a definition of the Y variable. It is defined later in the methods, but it would be helpful to define it at first use.

The legend for Figure 1a and b should be referring to Methods. There are too many non-obvious definitions requiring cross-checking the text.

p.1 l.54 - when citing scHi-C studies, citing most recent studies, e.g., Ulianov et al., "Order and Stochasticity in the Folding of Individual Drosophila Genomes.", Tan et al., "Changes in Genome Architecture and Transcriptional Dynamics Progress Independently of Sensory Experience during Post-Natal Brain Development.", may be considered.

p.2 l.62 - the statement that most recent approaches lack a generative model component may be reconsidered in light of work of Highsmith and Cheng, "VEHiCLE: a Variationally Encoded Hi-C Loss Enhancement algorithm for improving and generating Hi-C data" that uses GAN and describes previous methods.

Fig S14 and other supplementary figure legends have [?] - citations missing

I read the preprint, and the deeper dive for reviewing did not disappoint. I could notice only a handful of places requiring corrections, but overall got positive impression.

Reviewer 2

Were you able to assess all statistics in the manuscript, including the appropriateness of statistical tests used? No.

Were you able to directly test the methods? No.

Comments to author:

In the article entitled "Normalization and De-noising of Single-cell Hi-C Data with BandNorm and 3DVI", Zheng et al. present a normalization approach named BandNorm without batch correction and a deep generative model based method called 3DVI to account for technical noise and bias including batch effect for single cell Hi-C data. They evaluated these two methods together with other existing methods on four datasets in terms of specificity on distinguishing different cell types, robustness to technical noise and bias especially batch effect when doing clustering and impact on downstream analysis. In addition to these two methods, they also proposed an scGAD score for single cell Hi-C data, which borrowed the GAD concept from bulk Hi-C data analysis to find the cell type specific marker genes from single cell Hi-C data to facilitate functional interpretation and annotation of single cell Hi-C data.

Although these two methods showed comparable or superior performance on some of datasets in some of evaluation matrices compared to existing methods, there are three major problems with this work: i), the comparison is questionable and not systematic, thus part of the current result and conclusions need re-evaluate under more systematic and specific comparisons. ii) the methods of BandNorm and 3DVI seems too simple or lack of novelty. While BandNorm is a very simple algorithm to normalize Hi-C data by considering the distance effect and library size, 3DVI in essence is simply based on the framework of scVI-tools without much explicit modifications. iii) the presentation of the manuscript caused a lot of confusion to this reviewer. I do not see the points of present three different ideas in one paper. Although BandNorm+Harmony and 3DVI seem standing out from other methods after a bunch of

comparisons using different datasets under different matrices, this reviewer still wonders which strategy to choose for de-noising and integrative analysis of scHi-C data, e.g., BandNorm+Harmony or 3DVI. The paper is not well organized by conclusions, but rather by simply listing the result from many evaluation matrices, which may reflect a lack of thorough thought on the task and their findings.

Major points:

1, It seems that the authors directly used scVI-tools without explicit modifications or any adjustment for scHi-C data, yet they call it 3DVI. This seems not acceptable. Moreover, did the authors rigorously try different hyperparameters to optimize the model for a better performance over BandNorm + Harmony?

2, An ablation study that does not model batch factor into the 3DVI is required to illustrate the batch effect correction is fulfilled by 3DVI model design.

3, The comparison of normalization methods is not systematic and lack of detailed description. The normalization method BandNorm does not account for batch effect (as demonstrated in Figure 4) and needs additional step to correct for batch effect (with Harmony performed well for BandNorm). How are the embeddings in Figure 2 and 3 generated for those BandScale, CellScale and BandScale+CNN? Were they generated after batch correction? Moreover, why only applied CNN to BandScale to account for matrix structure of data but not to BandNorm and CellScale? Will CNN help to learn a better representation after different normalization?

4, The authors used the ARI score and the average Silhouette score to assess the performance of cell-type separation of different methods, however, the performance of batch effect correction of different methods also need to be evaluated using scores such as the batch entropy mixing score (doi:10.1038/nbt.4091, 2018) or the LISI scores (doi:10.1038/s41592-019-0619-0, 2019).

5, following #3, since the normalization methods compared here (e.g., BandNorm) need further batch correction, it is not suitable to compare the normalization methods followed by a batch correction with other methods, which confounds the superior performance of BandNorm (whether it is due to BandNorm or the batch removal methods). An unbiased or clear way could be comparing these methods in a batch-free context.

6, In Fig 3a, and in Fig 4 panel 'None', it seems that BandNorm cannot separate different cell types well without Harmony. L2/3, L5, and L6 are mixed together, Sst and Vip are mixed together. It is very likely that it is Harmony but not the BandNorm normalization that separates different cell types. This reviewer wonders that the authors should also combine other methods with Harmony to assess cell types separation performance. Again, all comparisons should be done under the same standard.

7, In the analysis of Figure 4, the results of 3DVI are not included and the benchmarking of batch effect correction seems superficial. The authors should perform rigorous benchmarking analysis to prove that 3DVI outperforms existing batch correction methods that may be originally developed for other types of single cell data but can be applied to scHi-C data directly. See <https://doi.org/10.1186/s13059-019-1850-9> where some scRNA-seq integration tools are

compared.

8, Why scHiC Topics is not included in the comparison in terms of impact on downstream analysis?

9, Although the overall similarity between aggregated scHi-C matrices and the corresponding bulk Hi-C matrices of 3DVI is high, the off-diagonal interactions (long-range) are almost missing in 3DVI, which is inspected from Figure 6 and 7. The authors should discuss and explain this.

10, In discussion, the authors suggested "3DVI ... provide advantages for downstream analysis compared to BandNorm", however, they used BandNorm normalized matrix for scGAD score calculations and UMAP analysis. Why?

11, this reviewer is curious about the overall correlation between scGAD scores and gene expression. The authors only demonstrated the scGAD scores on one dataset with paired scHi-C data and gene expression data measured by MALBAC-DT. This reviewer encourages the authors to perform more rigorous tests on other datasets.

Minor points:

1, Page 4 Line25-32: "All the methods except CellScale achieve their best performances on the Kim2019 dataset (ARI 2 [0.35, 0.91] and Silhouette score 2 [0.3, 0.73]), with leading performances by BandNorm, Higashi, scHiC Topics, and 3DVI (Fig. 1c)". The "Kim2019" should be "Kim2020"?

2, In Table 2, the authors should include the memory usage information.

Below are the point-by-point responses to the concerns for reviewer 1:

The manuscript "Normalization and De-noising of Single-cell Hi-C Data with BandNorm and 3DVI" presents two single-cell Hi-C data analysis methods. BandNorm is a normalization approach, and 3DVI is a deep generative modeling framework utilizing Poisson and Negative Binomial distributions to model scHi-C counts accounting for library size and batch effect for each band matrix to minimize technical biases and denoise scHi-C data. It also uses gene-centric analysis of scHi-C data by adapting the concept of single-cell gene-body associating domain (scGAD) scores. The methods were benchmarked on four public scHi-C datasets using the band-specific scaling (BandScale), the global scaling (CellScale), BandScale+CNN, scHiCluster, scHiC Topics, Higashi. Cell type separation, addressing batch effect, the effect of sequencing depth/sparsity, several metrics were evaluated. Investigation of the impact of normalization and denoising on downstream analysis, concordance in A/B

compartments, TAD boundaries (insulation score), agreement between different cell types, agreement between differential TAD boundaries and differential interactions. Single-cell gene-body associating domain (scGAD) score analysis allows for gene-level analysis and supports novel clustering enabling by the BandNorm normalization. The manuscript contains lots of information supported by figures and methods.

Bands' definition is unclear. Each subsequent diagonal band will be shorter than the previous and will be just a scalar at D. Shorter vectors will be more affected by outliers, especially pronounced in farther off-diagonal vectors. A similar approach, multiHiCcompare, progressively combines bands farther off-diagonal, which they dubbed 'progressive pooling'. How the problem of outliers, especially at larger distances, has been addressed/investigated, and whether a 'progressive pooling' strategy is warranted?

We thank the reviewer for this excellent suggestion. Indeed, the farther off-diagonal regions correspond to ultra-long-range interactions that tend to be highly sparse, and fewer cells have nonzero interactions in those regions. Our first version of scVI-3D software relied on setting the upper bound for the band indices to remove the last five to ten unstable bands where scVI-3D may fail due to lack of interactions. We are excited to find that the progressive pooling, which combines the second and third off-diagonals, fourth to sixth, seventh to tenth bands, and so on, works extremely well for scVI-3D. Additional file 1: Fig. S3 demonstrated that with the progressive pooling, scVI-3D achieves a markedly improved cell type separation between the excitatory neuronal subtypes (L2/3, L4, L5, L6) and inhibitory cells (Ndnf, Vip, Pvalb, and Sst), as well as stronger segregation between OPC and Astrocyte cells. The ARI and Silhouette scores also confirmed the UMAP visualization regarding the cell type separation. In addition, progressive pooling reduced the batch factor impact as the Batch ARI went from 0.06 to 0.04. Another key advantage that progressive pooling yielded for scVI-3D is the computational speed. Progressive pooling significantly reduced the run time due to fewer band matrices to model, and the updated version of scVI-3D is able to process the whole Lee2019 data set within 20 min with GPU and 23 CPU cores instead of 4 hours in our previous version. After comparing five different pooling strategies, the progressive pooling strategy introduced in multiHiCcompare stood out as the best performing and has been incorporated into the scVI-3D pipeline.

We also considered a number of progressive pooling strategies for BandNorm. However, Additional file 1: Fig. S34 indicated that the genomic distance decay profile, which is the major component of BandNorm normalization, is affected by the pooling procedure and becomes less precise. Consequently, the cell type separation performance is worsened according to Additional file 1: Fig. S3. Therefore, we only employ the progressive pooling strategy in scVI-3D.

p.3 l.48 - when citing the BNBC method, two references seem missing. The BNBC paper itself refers to the HiCcompare paper [PMID: 30064362] which introduces distance-focused normalization. And, the aforementioned multiHiCcompare paper explicitly used bands for normalization and differential analysis.

We appreciate the reviewer for providing additional literature support for the band-wise normalization. We have added both citations to places where we first introduced band transformation and where we started discussing the band-wise normalization for scHi-C data:

“Therefore, we first describe a computationally fast scaling normalization approach, named BandNorm (Fig. 1a), that operates on the stratified off-diagonals (i.e., bands) of the contact matrix and its variants as fast baseline alternatives, namely CellScale and BandScale which have been utilized for bulk Hi-C and have seen some uptake for scHi-C [10, 23, 24] (Methods).

.....

In the former category, in addition to BandNorm, we devised and evaluated two more baseline scaling-based normalization methods: CellScale and BandScale [10, 23, 24] (Methods).

.....

BandScale uses band-specific size factors rather than a global size factor within a cell and has been used previously to eliminate library size bias at each genomic distance [10, 23, 24].”

Equation 1 needs a definition of the Y variable. It is defined later in the methods, but it would be helpful to define it at first use.

Thanks for pointing this out. We moved the detailed definition in the “Band transformation of scHi-C data” sub-section before introducing equation 1 in the Methods section.

The legend for Figure 1a and b should be referring to Methods. There are too many non-obvious definitions requiring cross-checking the text.

We appended the legends with a note referring to Methods for model details.

p.1 l.54 - when citing scHi-C studies, citing most recent studies, e.g., Ulianov et al., "Order and Stochasticity in the Folding of Individual *Drosophila* Genomes.", Tan et al., "Changes in Genome Architecture and Transcriptional Dynamics Progress Independently of Sensory Experience during Post-Natal Brain Development.", may be considered.

Thanks for the suggestions, we have now cited these:

“Maturation of chromosome conformation capture (3C) based technologies for profiling 3D genome organization [1, 2, 3, 4, 5] and technological advancements in single-cell sequencing [6] led to the development of single-cell Hi-C (scHi-C) assays [7, 8, 9, 10, 11, 12].”

p.2 l.62 - the statement that most recent approaches lack a generative model component may be reconsidered in light of work of Highsmith and Cheng, "VEHiCLE: a Variationally Encoded Hi-C Loss Enhancement algorithm for improving and generating Hi-C data" that uses GAN and describes previous methods.

We thank the reviewer for making us aware of this method. It adds more evidence to the background discussion about utilizing the generative model for efficient and scalable normalization, de-noising, and imputation of bulk and other single-cell modality analysis. However, we note that the generative model constructed in VEHiCLE is still designed for the bulk Hi-C data instead of the scHi-C data. Therefore, more investigations are needed to understand the performance of GAN in single-cell Hi-C studies. We cited VEHiCLE as below:

“While these are highly innovative approaches, they lack a generative model that acknowledges the key properties of the scHi-C data. Deep generative modeling and, more specifically, variational autoencoders have seen a significant uptake in the analysis of single-cell transcriptomics [18, 19], epigenomics [20], and proteomics [21] and bulk 3D genomics analysis [22] due to their ability to provide efficient and scalable solutions to normalize, de-noise, and impute single-cell data.”

Fig S14 and other supplementary figure legends have [?] - citations missing

We apologize for overlooking this issue. The citations are properly added.

I read the preprint, and the deeper dive for reviewing did not disappoint. I could notice only a handful of places requiring corrections, but overall got positive impression.

We appreciate the reviewer’s positive feedback and reemphasize that we are grateful for the progressive pooling suggestion.

Below are the point-by-point responses to the major concerns for reviewer 2:

In the article entitled "Normalization and De-noising of Single-cell Hi-C Data with BandNorm and 3DVI", Zheng et al. present a normalization approach named BandNorm without batch correction and a deep generative model based method called 3DVI to account for technical noise and bias including batch effect for single cell Hi-C data. They evaluated these two methods together with other existing methods on four datasets in terms of specificity on distinguishing different cell types, robustness to technical noise and bias especially batch effect when doing clustering and impact on downstream analysis. In addition to these two methods, they also proposed an scGAD score for single cell Hi-C data, which borrowed the GAD concept from bulk Hi-C data analysis to find

the cell type specific marker genes from single cell Hi-C data to facilitate functional interpretation and annotation of single cell Hi-C data.

Although these two methods showed comparable or superior performance on some of datasets in some of evaluation matrices compared to existing methods, there are three major problems with this work: i), the comparison is questionable and not systematic, thus part of the current result and conclusions need re-evaluate under more systematic and specific comparisons. ii) the methods of BandNorm and 3DVI seems too simple or lack of novelty. While BandNorm is a very simple algorithm to normalize Hi-C data by considering the distance effect and library size, 3DVI in essence is simply based on the framework of scVI-tools without much explicit modifications. iii) the presentation of the manuscript caused a lot of confusion to this reviewer. I do not see the points of present three different ideas in one paper. Although BandNorm+Harmony and 3DVI seem standing out from other methods after a bunch of comparisons using different datasets under different matrices, this reviewer still wonders which strategy to choose for de-noising and integrative analysis of scHi-C data, e.g., BandNorm+Harmony or 3DVI. The paper is not well organized by conclusions, but rather by simply listing the result from many evaluation matrices, which may reflect a lack of thorough thought on the task and their findings.

We thank the reviewer for the thoughtful comments that prompted further evaluations of our methods. We extended our computational experiments and paid more attention to highlighting broader conclusions of our experiments to guide the users about the usage of BandNorm and scVI-3D.

Major points:

1. It seems that the authors directly used scVI-tools without explicit modifications or any adjustment for scHi-C data, yet they call it 3DVI. This seems not acceptable.

We appreciate the reviewer's sensitivity to this. We would like to explicitly point out our novel contributions in setting up the conceptual framework and implementation details. First, 3DVI goes well beyond a naive application of scVI on scHi-C data because scVI requires a (genes) by (cells) matrix as input and it cannot be directly fit on the (locus-pair) x (locus-pair) matrices of the cells. In fact, we considered a number of alternatives before we arrived at the band transformation. Most notably, we vectorized the (locus-pair) x (locus-pair) as we have done for BandNorm and inputted this into scVI; however, this formulation failed because of the systematic band decay bias in scHi-C data. We attempted to bypass this by explicitly including the band decay as a covariate; however, this was not sufficient enough to correct the apparent bias. Ultimately, we utilized the band transformation in BandNorm to obtain the (locus-pair) x (cell) matrices per band. Such a band transformation bridges scHi-C data with the normalization and dimension reduction methods from other single-cell modalities.

In the analysis of scRNA-seq data, genes (features) are extensively filtered for low variability, e.g., a typical analysis keeps about 2-5K genes out of 30-50K. The filtering procedure for scHi-C data differs significantly from scRNA-seq data. We keep all the features (i.e., locus-pairs) on each band matrix and only filter each band matrix to exclude the cells with no interaction across all the locus-

pairs within each band. Then, the variational inference analysis is sequentially applied to each band matrix. Consequently, the latent embeddings on such band matrices are missing for the no interaction cells. To concatenate latent embeddings from different band matrices while matching the cell identity, we explored a series of latent embedding imputation approaches, such as convolutional neighborhood imputation and averaging across all the cells on each feature. Eventually, we found that imputing by 0 achieved the best performance. Additionally, we also explored the impact of the dimension of latent space on the performance and found that latent space dimension around 100 generally works the best for scHi-C data while scRNA-seq usually only need an order of fewer (e.g., 10-50) latent variables (Additional file 1: Fig. S4). Furthermore, we incorporated a progressive pooling strategy to aggregate the farther off-diagonals in a progressive manner to pool sufficient interactions and alleviate the sparsity issues of scHi-C data (Methods and Additional file 11: Fig. S3).

We included more implementation details in our manuscript to highlight that 3DVI is not an off-the-shelf application of scVI. We have also changed the method name from 3DVI to scVI-3D to better acknowledge the framework constructed by the authors of scVI.

Moreover, did the authors rigorously try different hyperparameters to optimize the model for a better performance over BandNorm + Harmony?

This is a great point to raise, and, in fact, we did. When we realized BandNorm was performing better despite its simplicity, we scrutinized scVI-3D thoroughly. Applying scVI-3D with default parameters performs poorly, which is perhaps not so surprising as the sparsity structure of these two data types (scRNA-seq vs. scHi-C) is vastly different. We optimized these parameters so that the new defaults under our adaptation would perform well with scHi-C data. For example, Additional file 1: Fig. S4 demonstrates an optimal latent space dimension being around 100, which we set as the default parameter value.

2. An ablation study that does not model batch factor into the 3DVI is required to illustrate the batch effect correction is fulfilled by 3DVI model design.

Thanks for the suggestion. We added quantitative assessment and visual comparisons between scVI-3D with and without modeling of the batch effects. Additional file 1: Fig. S15 confirmed the decreased batch impact and improvement in the sub-neuron cell types separation. Adjusted rand index evaluating the batch effect is also lower when scVI-3D implements the batch correction. This observation is further supported by the integration local inverse Simpson's Index (LISI; Korsunsky et al. 2019. Nature Methods), which also shows depletion of cell enrichment around an index score of 1. There is also overwhelming evidence in the literature that the general modeling strategy of scVI provides a competitive way of removing the batch effects. For example, recent experiments of Lakkis et al. (Genome Research, 2021) confirmed that their method CarDEC and scVI not only removed the multi-level batch effects but also preserved inter-cell type variation.

3. The comparison of normalization methods is not systematic and lack of detailed description. The normalization method BandNorm does not account for batch effect (as demonstrated in Figure 4) and needs additional step to correct for batch effect (with Harmony performed well for BandNorm). How are the embeddings in Figure 2 and 3 generated for those BandScale, CellScale and BandScale+CNN? Were they generated after batch correction?

Analysis of Ramani2017 and Kim2020 does not require batch correction because Ramani2017 does not exhibit batch effects, and the experimental design in Kim2020 confounds batches and cell types, where each cell type is either in a single batch alone or together with another cell line in two batches. Therefore, we focused our batch correction discussion on the Lee2019 data set. In Fig. 3, all the methods that do not explicitly model the batch effects, including CellScale, BandScale, BandNorm, scHiC Topics, and CellScale+CNN, are coupled with Harmony to perform batch correction while scHiCluster, Higashi, and scVI-3D leverage their built-in batch removal. UMAP visualization and quantitative evaluation in Fig. 3 illustrate that while scHiC Topics still exhibits some lingering batch effects, all other methods are able to remove the batch effects successfully.

Moreover, why only applied CNN to BandScale to account for matrix structure of data but not to BandNorm and CellScale? Will CNN help to learn a better representation after different normalization?

We appreciate the reviewer for bringing up further discussion on CNN as we did, in fact, evaluate BandNorm+CNN while we were preparing the first version of our manuscript. This combination performed poorly, barely separating the major cell types for the Ramani2017 and Kim2020 studies. With a more comprehensive comparison between CellScale+CNN, BandScale+CNN, and BandNorm+CNN, both visually and quantitatively, we found that CellScale+CNN has the most outstanding performance among those three (Additional file 1: Fig. S6). Therefore, we included CellScale+CNN in the downstream systematic benchmarking across eight scHi-C normalization and denoising methods. However, the performance of CellScale+CNN is still not competitive, ranking 6th over four data sets and six evaluation metrics (Fig. 1d). We attributed this to the low resolution and high sparsity. At 1Mb, the graph structure of each cell matrix has limited information, hence, acknowledging the matrix structure of the data does not lead to discernable gains in cell type separation power.

4. The authors used the ARI score and the average Silhouette score to assess the performance of cell-type separation of different methods, however, the performance of batch effect correction of different methods also need to be evaluated using scores such as the batch entropy mixing score (doi:10.1038/nbt.4091, 2018) or the LISI scores (doi:10.1038/s41592-019-0619-0, 2019).

Thanks for the great suggestions. Our preliminary analysis indicates that integration LISI scores generate more distinguishable results between scHi-C data with and without batch effect than the entropy mixing scores, which tend to be zero in all the methods and testing scenarios. The adjusted rand index (ARI) is also widely used in literature to measure the batch effect by measuring the consistency between the clustering label with the batch label, as demonstrated,

For example, in Tran et al. (Genome biology 2020) and Lin et al. (Proceedings of the National Academy of Sciences 2019). Therefore, we leveraged the integration LISI scores and ARI to measure the impact of the batch factor (Fig. 3, Additional file 1: Figs. S3-5 and S15-16).

5-6. Following #3, since the normalization methods compared here (e.g., BandNorm) need further batch correction, it is not suitable to compare the normalization methods followed by a batch correction with other methods, which confounds the superior performance of BandNorm (whether it is due to BandNorm or the batch removal methods). An unbiased or clear way could be comparing these methods in a batch-free context.

In Fig 3a, and in Fig 4 panel 'None', it seems that BandNorm cannot separate different cell types well without Harmony. L2/3, L5, and L6 are mixed together, Sst and Vip are mixed together. It is very likely that it is Harmony but not the BandNorm normalization that separates different cell types. This reviewer wonders that the authors should also combine other methods with Harmony to assess cell types separation performance. Again, all comparisons should be done under the same standard.

We thank the reviewer for pointing out the potential confounding effect between the cell type separation performance of BandNorm and the batch correction of Harmony. We first looked into the cell type separation performance of BandNorm without Harmony correction on all the cell types and excitatory and inhibitory sub-neuron cells (L2/3, L4, L5, L6, Ndnf, Pvalb, Sst, and Vip), respectively. Additional file 1: Fig. S16B demonstrates that BandNorm normalization already achieves a solid cell-type separation across all the cells assessed by the ARI and Silhouette scores. The Batch ARI improves further with Harmony batch effect correction, but there is no dominating pattern showing that the cell type separation metrics are consistently higher across all the cell types. The improvement is more dramatic among the sub-neuron cell types, which agrees with the UMAP visualization in Additional file 1: Fig. S16A.

Next, we adapted Harmony as part of the BandNorm R function to correct batch effects when needed. As the reviewer suggested, Harmony has also been added to all the methods that do not have an inherent batch correction, including CellScale, BandScale, BandNorm, CellScale+CNN, and scHiC Topics. Fig. 3 demonstrates that BandNorm, Higashi, and scHiCluster are the leading methods that can successfully separate the excitatory and inhibitory sub-cell types of the Lee2019 data set. scVI-3D demonstrates a specific advantage in separating the sub-neuron cell types at high resolution compared to BandNorm (Additional file 1: Fig. S5). Except for scHiC Topics, all the methods can sufficiently address the batch effect (Fig. 3c-f). Furthermore, we also leveraged a single library (190315_21yr in Lee2019 data) to create a batch-free setting for benchmarking the cell type separation performances (Additional file 1: Fig. S14). BandNorm, Higashi, scHiCluster, and scVI-3D outperformed the other methods in this batch-free setting as well.

7. In the analysis of Figure 4, the results of 3DVI are not included and the benchmarking of batch effect correction seems superficial. The authors should perform rigorous benchmarking analysis to prove that 3DVI outperforms existing batch correction methods that may be originally developed for other types of single cell data but can be applied to scHi-C data directly. See

<https://doi.org/10.1186/s13059-019-1850-9> where some scRNA-seq integration tools are compared.

The original Figure 4, which is the current Additional file 1: Fig. S16A, focuses on investigating and seeking a proper batch correction method in combination with BandNorm normalization for scHi-C datasets that show severe batch effects, i.e., Lee2019 data set. We do not intend to propose a method or methods combination that can achieve the universally best performance in eliminating the batch effect compared to other candidate algorithms. Instead, we found Harmony can be a practical option for BandNorm and other scHi-C methods that do not explicitly model the batch effects. After we released our findings on bioRxiv, scHiCluster also chose to adapt Harmony into its framework for additional batch correction in their latest launched software. scVI-3D, on the other hand, adapts batch correction capabilities of scVI which has an established good performance for scRNA-seq and demonstrates expected successful batch effect removal on scHi-C data (Fig. 3 and Additional file 1: Fig. S15).

8. Why scHiC Topics is not included in the comparison in terms of impact on downstream analysis?

scHiC Topics took a long time to run (36hours) and was ranked as the fifth-best method in the benchmarking section. Therefore, we only considered the first four methods in the downstream analysis comparison.

9. Although the overall similarity between aggregated scHi-C matrices and the corresponding bulk Hi-C matrices of 3DVI is high, the off-diagonal interactions (long-range) are almost missing in 3DVI, which is inspected from Figure 6 and 7. The authors should discuss and explain this.

Thanks for pointing it out. The missing long-range interactions are due to the improper color scale. We adjusted the color scale for aggregated scHi-C matrices and the bulk matrices separately and replaced the corresponding figures with a more visible version (Figs. 5-6 and Additional file 1: Figs. 18-19).

10. In discussion, the authors suggested "3DVI ... provide advantages for downstream analysis compared to BandNorm", however, they used BandNorm normalized matrix for scGAD score calculations and UMAP analysis. Why?

We conclude that scVI-3D provides advantages over BandNorm when data are very sparse and have fewer cells (less than 50 cells), necessitating imputation. Also, the zero-inflation model enables scVI-3D to better handle the high-resolution data, i.e., Lee2019 at 100kb better than BandNorm. Overall, however, we still recommend BandNorm, which achieves the most stable and high-performance in the majority of the tasks investigated. Besides, BandNorm is much faster for large data. Tan2021 data set has the highest sequencing depth and genome coverage publicly released compared to other scHi-C assays. 100kb is an adequate resolution without running into severe high sparsity issues (Additional file 1: Fig. S23). Therefore, for the Tan2021 data set, which

has sufficient sequencing depth and relatively low sparsity level, BandNorm is more suitable for the analysis.

11. This reviewer is curious about the overall correlation between scGAD scores and gene expression. The authors only demonstrated the scGAD scores on one dataset with paired scHi-C data and gene expression data measured by MALBAC-DT. This reviewer encourages the authors to perform more rigorous tests on other datasets.

Thanks for this suggestion. We now included figures depicting the correlations between scGAD and gene expression more comprehensively using other scHi-C datasets. Apart from Tan et al. 2021 Dip-C data with MALBAC-DT, we also use scGAD with Paired-Tag data from Zhu et al. 2021 (<https://doi.org/10.1038/s41592-021-01060-3>) and Lee2019 with the matching human motor cortex scRNA-seq data released by the BRAIN Initiative Cell Census Network (Bakken et al. 2021, <https://doi.org/10.1038/s41586-021-03465-8>). The Pearson correlation coefficient is calculated across all the eligible genes for each cell type. We also added a correlation baseline through permuting the cell type labels as depicted as the gray bars in Additional file 1: Fig. S25. We also provided the corresponding scatter plots illustrating the relationship between scGAD and scRNA-seq gene expression in Additional file 1: Fig. S26-28. We found scGAD can bridge 3D genomics and transcriptomics for multimodality information sharing and summarized our findings in another paper currently available on BioRxiv (<https://www.biorxiv.org/content/10.1101/2021.10.22.465520v1>).

Minor points:

1. Page 4 Line 25-32: "All the methods except CellScale achieve their best performances on the Kim2019 dataset (ARI 2 [0.35, 0.91] and Silhouette score 2 [0.3, 0.73]), with leading performances by BandNorm, Higashi, scHiC Topics, and 3DVI (Fig. 1c)". The "Kim2019" should be "Kim2020"?

We apologize for the typo. We have fixed it.

2. In Table 2, the authors should include the memory usage information.

Thanks for the suggestion. We have added the memory usage in Table 2.

Second round of review

Reviewer 1

All comments have been addressed.

Reviewer 2

Manuscript by Zheng et al. has now been revised, and the authors have provided substantial new results to support their claims, including more analysis of confounding effects such as batch correction. They also performed more rigorous comparisons between BandNorm and other normalization methods. In addition, the paper was re-organized by conclusions, making it much more comprehensible. We appreciate the clarification about the methodological contributions of scVI-3D and its advantages over BandNorm under certain circumstances.

We have only a few small remaining concerns:

1) While the authors now apply the integration local inverse Simpson's Index (iLISI) scores to evaluate the performance of batch mixing, they do not describe in the methods section how this score is defined and calculated.

2) My last concern is code reproducibility. Could the authors deposit the data analysis code onto public website, e.g., GitHub?

Authors Response

Point-by-point responses to the reviewers' comments:

Reviewer #2 - 1) While the authors now apply the integration local inverse Simpson's Index (iLISI) scores to evaluate the performance of batch mixing, they do not describe in the methods section how this score is defined and calculated.

We thank the reviewer for pointing this out. We have added the introduction and calculation of iLISI in the Methods section.

"Integration Local Inverse Simpson's Index (iLISI) score. The iLISI [35] is employed with the "compute_lisi" function of the R package LISI to evaluate the batch effects. A high density of cells around one indicates that the neighborhoods of these cells have a single batch representation; hence, it signifies batch effects. In contrast, larger iLISI scores indicate that batches are well mixed in the cells' neighborhood, signaling a low impact of the batch effect."

Reviewer #2 - 2) My last concern is code reproducibility. Could the authors deposit the data analysis code onto public website, e.g., GitHub?

We thank the reviewer for the good suggestion. The data analysis codes are deposited to the GitHub (https://github.com/keleslab/BandNorm_and_scVI-3D_manuscript) and Zenodo (<https://doi.org/10.5281/zenodo.7084396>) repositories.